# Lysosomal TBK1 responds to amino acid availability to relieve Rab7-dependent mTORC1 inhibition

Gabriel Talaia [ID] [1,2,3,4,5], Amanda Bentley-DeSousa[1,2,3,4,5] & Shawn M Ferguson [ID] [1,2,3,4,5,6 ✉]

## Abstract

**Lysosomes play a pivotal role in coordinating macromolecule degradation and regulating cell growth and metabolism. Despite substantial progress in identifying lysosomal signaling proteins, understanding the pathways that synchronize lysosome functions with changing cellular demands remains incomplete. This study uncovers a role for TANK-binding kinase 1 (TBK1), well known for its role in innate immunity and organelle quality control, in modulating lysosomal responsiveness to nutrients. Specifically, we identify a pool of TBK1 that is recruited to lysosomes in response to elevated amino acid levels. This lysosomal TBK1 phosphorylates Rab7 on serine 72. This is critical for alleviating Rab7-mediated inhibition of amino acid-dependent mTORC1 activation. Furthermore, a TBK1 mutant (E696K) associated with amyotrophic lateral sclerosis and frontotemporal dementia constitutively accumulates at lysosomes, resulting in elevated Rab7 phosphorylation and increased mTORC1 activation. This data establishes the lysosome as a site of amino acid regulated TBK1 signaling that is crucial for efficient mTORC1 activation. This lysosomal pool of TBK1 has broader implications for lysosome homeostasis, and its dysregulation could contribute to the pathogenesis of ALS-FTD.**

**Keywords** Lysosome; Nutrient Sensing; mTORC1; ALS-FTD; TBK1
**Subject Categories** Autophagy & Cell Death; Membranes & Trafficking; Organelles

## Introduction

In addition to degrading macromolecules that are delivered by the endocytic and autophagy pathways, lysosomes act as signaling platforms that control cell growth and metabolism (Ballabio and Bonifacino, 2019; Ferguson, 2015; Goul et al, 2023). Coordinating the degradative and signaling functions of lysosomes is crucial for ensuring that cells efficiently recycle damaged or surplus components while precisely regulating their growth and energy use according to the cell's needs and external cues. Mechanistic target

of rapamycin complex 1 (mTORC1) is a kinase that communicates between lysosomes and the rest of the cell to help match growth to the availability of nutrients and growth factors (Goul et al, 2023; Liu and Sabatini, 2020). To this end, multiple nutrients are sensed and integrated upstream of the heterodimeric Rag GTPases that recruit mTORC1 to the surface of lysosomes (Goul et al, 2023; Lama-Sherpa et al, 2023; Liu and Sabatini, 2020). Meanwhile, growth factor signaling controls Rheb, a GTPase that is responsible for mTORC1 activation at lysosomes (Angarola and Ferguson, 2020; Menon et al, 2014). This complex coordination between diverse inputs to regulate signals from lysosomes highlights the critical need for cells to balance their degradative activities with overall nutrient requirements.

Cells also monitor and manage lysosome integrity to ensure that lysosomes remain intact and lysosomal enzymes do not leak into the cytoplasm (Bohannon and Hanson, 2020; Yang and Tan, 2023). When lysosomes suffer damage that cannot be repaired, TANK-binding kinase 1 (TBK1) helps cells to clear these severely damaged lysosomes via the process known as lysophagy where the damaged lysosomes are engulfed by autophagosomes and then delivered to intact lysosomes for degradation (Eapen et al, 2021; Hung et al, 2013; Maejima et al, 2013; Pied et al, 2022; Vargas et al, 2023). TBK1 also plays important roles in the autophagic clearance of damaged mitochondria and in signaling related to innate immunity (Fitzgerald et al, 2003; Harding et al, 2021; Heo et al, 2018; Moore and Holzbaur, 2016; Pomerantz and Baltimore, 1999). Heterozygous loss-of-function mutations in the *TBK1* gene cause amyotrophic lateral sclerosis (ALS) and frontotemporal dementia (FTD) (Cirulli et al, 2015; Freischmidt et al, 2015; Gijselinck et al, 2015). TBK1 missense mutations also cause ALS and FTD (Freischmidt et al, 2015; Gijselinck et al, 2015; Hirsch-Reinshagen et al, 2019; Pottier et al, 2015; Van Mossevelde et al, 2016). Heterozygous loss-of-function mutations in TBK1 also leads to herpes simplex encephalitis (Ahmad et al, 2016). Meanwhile, excess TBK1 activity arising from gene duplication causes open angle glaucoma (Ahmad et al, 2016). Aberrant TBK1 activity can also contribute to diabetes and obesity (Bodur et al, 2022; Cruz et al, 2018; Oral et al, 2017; Reilly et al, 2013). These diseases arising from both too little and too much TBK1 activity demonstrate the importance of regulatory mechanisms that ensure tight control of TBK1 kinase activity across diverse cell types. This breadth of physiological and pathophysiological functions for TBK1 raises

[1]Department of Cell Biology, Yale University School of Medicine, New Haven, CT 06510, USA. [2]Department of Neuroscience, Yale University School of Medicine, New Haven, CT 06510, USA. [3]Program in Cellular Neuroscience, Neurodegeneration and Repair, Yale University School of Medicine, New Haven, CT 06510, USA. [4]Wu Tsai Institute, Yale University School of Medicine, New Haven, CT 06510, USA. [5]Aligning Science Across Parkinson's (ASAP) Collaborative Research Network, Chevy Chase, MD 20815, USA. [6]Kavli Institute for Neuroscience, Yale University School of Medicine, New Haven, CT 06510, USA. ✉E-mail: shawn.ferguson@yale.edu

questions about mechanisms that control the TBK1 activity involved in distinct processes at diverse subcellular sites.

In addition to substrates directly related to innate immunity and organelle quality control, TBK1 phosphorylates mTORC1 components leading to both activation and inhibition of mTORC1 kinase activity in different studies (Antonia et al, 2019; Bodur et al, 2018; Cooper et al, 2017; Hasan et al, 2017; Ye et al, 2023). These reports of opposing effects of TBK1 on mTORC1 activity are puzzling and raise questions about underlying mechanisms. As mTORC1 activation is now well known to take place at lysosomes, if TBK1 directly regulates mTORC1 or its immediate upstream regulators, then TBK1 should also be present on intact lysosomes and not just recruited in response to severe damage that induces lysophagy (Eapen et al, 2021; Goul et al, 2023). If so, there should also be signals that control TBK1 at intact lysosomes and lysosome-localized TBK1 substrates that mediate communication from TBK1 to mTORC1.

To investigate lysosomal functions of TBK1 and to define the relationship between TBK1 and mTORC1 activation, we employed a series of genetic and pharmacological perturbations in mammalian cells in culture combined with assays of TBK1 subcellular localization, substrate phosphorylation, and mTORC1 activity. We observed that both acute pharmacological inhibition and genetic depletion of TBK1 resulted in a significant reduction in the ability of mTORC1 to be efficiently activated in response to feeding cells with amino acids. This need for TBK1 in mTORC1 activation was paralleled by observations that TBK1 is recruited to the surface of lysosomes concurrent with mTORC1 activation when starved cells are re-fed with amino acids. We furthermore established that TBK1-dependent phosphorylation of Rab7 (serine 72) is critical for relieving inhibition of mTORC1 by Rab7. Finally, we discovered that the ALS-FTD TBK1-E696K mutant is constitutively localized to lysosomes, is more active and more strongly promotes mTORC1 activation by amino acids. Altogether, this research demonstrates that amino acid availability controls the ability of a lysosomal pool of TBK1 to relieve Rab7-dependent inhibition of mTORC1 activation.

## Results

### TBK1 promotes amino acid-dependent activation of mTORC1 at lysosomes

To test for TBK1 functions upstream of mTORC1 at lysosomes, we performed genome editing to knock out TBK1 in HeLa cells and measured the impact on mTORC1 activation in assays where starved cells were re-fed with amino acids. These experiments revealed reduced mTORC1 activation by amino acids in the TBK1 KO cells as measured by phosphorylation of its direct substrates, S6K1 (phospho-T389) and ULK1 (phospho-S757; Figs. 1A,B and EV1A,B) (Liu and Sabatini, 2020). Similar results were obtained from mouse RAW264.7 macrophage cells when we knocked out both TBK1 and the closely related IKKε that is co-expressed with TBK1 in myeloid cells (Fig. EV1C,D) (Balka et al, 2020). To rule out long-term adaptation to the TBK1 KO as a contributing factor to our observations, we performed acute treatment of RAW264.7 cells with BX-795, an inhibitor of TBK1 kinase activity, and this yielded a similar effect to TBK1 KO on

mTORC1 signaling (Fig. 1C,D) (Bain et al, 2007). Furthermore, the specificity of the mTORC1 activation defect in TBK1 KO HeLa cells was demonstrated by the full rescue of this phenotype following stable expression of TBK1-GFP in the TBK1 KO cells (Fig. 1E,F). Confocal immunofluorescence microscopy revealed the presence of occasional TBK1 puncta adjacent to lysosomes in wild-type HeLa cells and this was lost in the TBK1 KO cells. This result was consistent with the possible localization of a sub-population at lysosomes that encouraged follow-up experiments to test this idea more rigorously. In the TBK1 KO cells that were rescued with TBK1-GFP at ~4x above endogenous levels, the lysosome localization of TBK1 was more evident (Fig. 1G). To test whether the partial TBK1 lysosome localization observed by immunofluorescence was coincidental, the presence of TBK1 at lysosomes was independently assessed by immunoblotting of lysosomes that were magnetically isolated from cells whose lysosomes were endocytically loaded with superparamagnetic iron oxide nanoparticles (SPIONs) via a pulse-chase strategy that we have previously validated to yield highly enriched lysosomes (Amick et al, 2018; Rodriguez-Paris et al, 1993). Both endogenous TBK1 and TBK1-GFP were present on the purified lysosomes along with established late endosome/lysosome proteins [LAMP1 and Rab7A (henceforth referred to as Rab7 for simplicity)] (Fig. 1H). As expected, PDI and GM130, proteins of the endoplasmic reticulum and Golgi, were depleted from the purified lysosome samples. Rab7, a small GTPase that regulates multiple aspects of late endosomes and lysosomes, was also abundant in the purified lysosomes (Amick et al, 2018; Kummel et al, 2023) (Fig. 1H). Rab7 has been reported to be a major substrate of TBK1 but the endolysosomal significance of this has not been extensively studied (Heo et al, 2018; Ritter et al, 2020). Interestingly, Rab7 phosphorylated on serine 72 (S72), the previously established TBK1 site, was detected in both the whole-cell lysates and the purified lysosomes (Fig. 1H). The abundance of this phospho-Rab7 in both fractions was reduced in the TBK1 KO cells and increased in the TBK1-GFP rescue line (Fig. 1H). To ensure that this TBK1-GFP fusion protein supports normal TBK1 regulation and functions, we treated TBK1 KO cells that express TBK1-GFP with cGAMP [an agonist for Stimulator of Interferon Genes (STING), a robust activator of TBK1] and confirmed that this resulted in the expected TBK1 autophosphorylation on S172, an increased interaction between TBK1 and STING and a band shift for STING that reflects its phosphorylation by TBK1 (Fig. EV1E) (Tanaka and Chen, 2012).

### TBK1 signaling at lysosomes is stimulated when amino acids are abundant

TBK1-dependent phosphorylation of Rab7 under basal cell growth conditions was quantified in whole-cell lysates from wild type, TBK1 KO, and TBK1-GFP rescued HeLa cells. Rab7 phosphorylation was decreased in the TBK1 KO cells and was rescued to above wild-type levels following the stable over-expression of TBK1-GFP in the TBK1 KO background (Fig. 2A,B). Phosphorylation of Rab7 by TBK1 was previously described to occur during the very specific contexts of mitophagy and innate immunity signaling in the STING pathway (Heo et al, 2018; Ritter et al, 2020). Our observations of TBK1 at lysosomes with impacts on Rab7 phosphorylation and mTORC1 activation under basal cell growth conditions in the absence of stimuli related to mitochondrial damage or STING

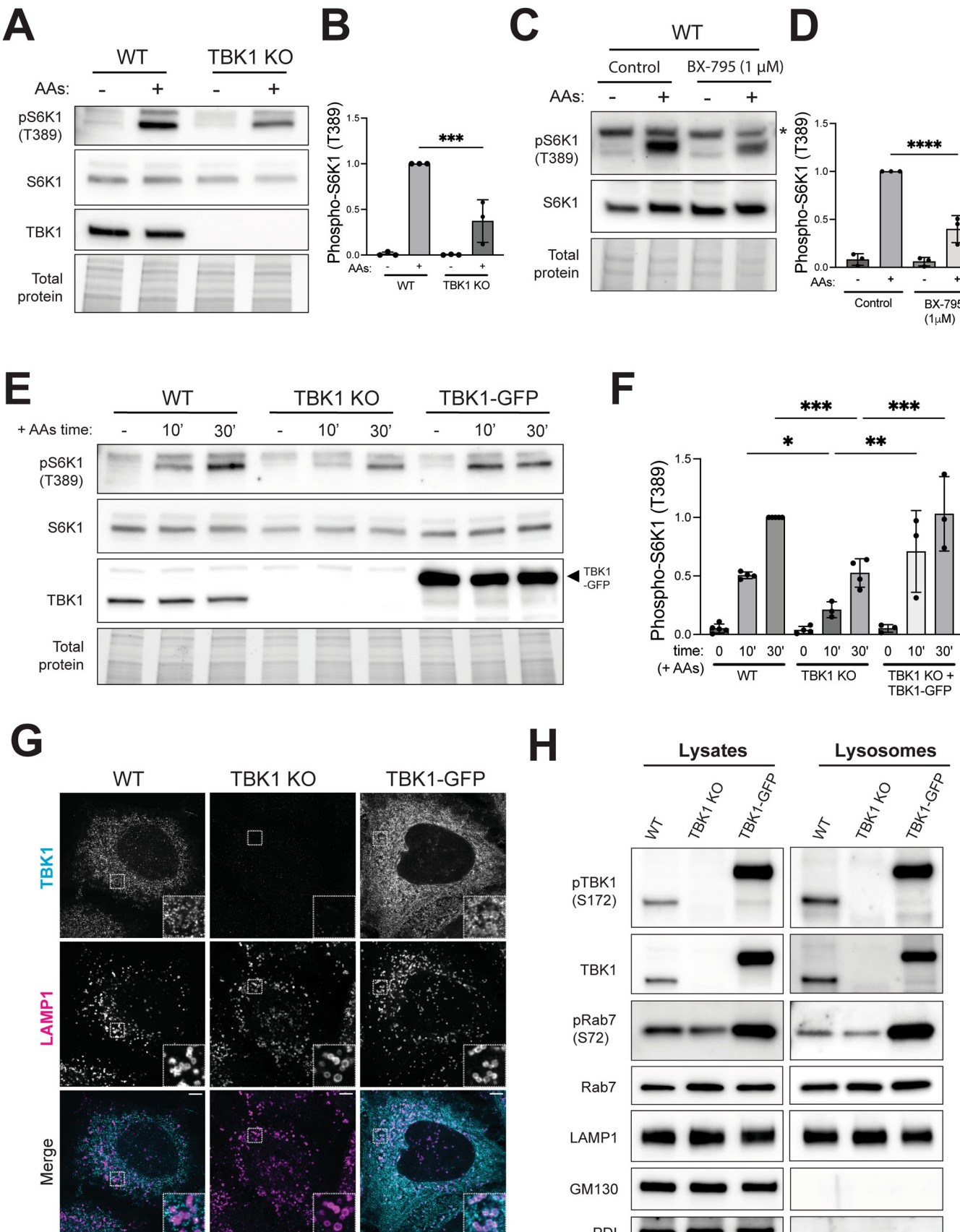

**Figure 1. Lysosomal TBK1 promotes amino acid-dependent mTORC1 signaling.**

(A) Immunoblot analysis of S6K1 phosphorylated at Thr389 [pS6K1 (T389)], total S6K1 and TBK1 in whole-cell lysates from WT versus TBK1 KO HeLa cells that were starved of amino acids for 60 min (−) and then re-fed with amino acids for 60 min (+). The TGX stain-free method was used to confirm similar total protein levels in each sample. (B) Quantification of phospho-S6K1 (T389) normalized to total S6K1 and expressed as a fold change compared to the WT cells under re-fed conditions. Data reflects three biological replicates and statistical significance was determined by ordinary one-way analysis of variance (ANOVA) with Šidák post hoc test ($n = 3$; mean ± SD; ***$p = 0.0002$). (C) Immunoblot analysis of RAW264.7 cells that underwent amino acid starvation (60 min) and re-feeding (60 min) in the presence of 1 µM BX-795 or 0.01% (v/v) DMSO (Control). The asterisk notes the position of a non-specific band that runs slightly above the phospho-S6K1 signal. (D) Quantification of phospho-S6K1 (T389) normalized to total S6K1 abundance and expressed as a fold change compared to the WT cells under re-fed conditions. Data reflects three biological replicates and statistical significance was determined by ordinary ANOVA with Šidák post hoc test ($n = 3$; mean ± SD; ****$p < 0.0001$). (E) Immunoblots from wild type (WT), TBK1 KO, and TBK1 KO stably expressing TBK1-GFP (TBK1-GFP) HeLa cells that were starved for 60 min (−) and then re-fed with amino acids for 10 or 30 min (10′ and 30′). The TGX stain-free method was used as a loading control (Total protein). (F) Phospho-S6K1 levels were quantified and normalized to total S6K1 from at least three biological replicates and expressed as fold change compared to the WT cells under 30 min re-fed conditions. Data reflects at least three biological replicates and statistical significance was determined by ordinary ANOVA with Šidák post hoc (mean ± SD; *$p = 0.0497$; ***$p = 0.0002$; **$p = 0.0010$; ***$p = 0.0004$). (G) Super-resolution spinning disk confocal immunofluorescence microscopy of TBK1 and LAMP1 in wild-type, TBK1 KO, and TBK1-GFP HeLa cells under basal growth conditions. Insets are magnified 3X. Scale bar: 5 µm. (H) Immunoblot analysis of the indicated proteins in cell lysates (Lysates) and magnetically isolated lysosomes (Lysosomes) from WT, TBK1 KO, and TBK1-GFP HeLa cells under basal growth conditions. Blots for Lysate and Lysosome fractions were obtained in parallel from the same membrane. Source data are available online for this figure.

activation raised two questions. First, what regulates this lysosomal TBK1? Second, does Rab7 phosphorylation contribute to TBK1-dependent activation of mTORC1? Answering these questions was important for establishing a functional impact for TBK1 at lysosomes.

Given that TBK1 is required for efficient mTORC1 activation in response to the addition of amino acids, we tested whether TBK1 activity is also regulated by this stimulus. Indeed, Rab7 phosphorylation increased when starved RAW264.7 cells were re-fed with amino acids and both basal and amino acid-stimulated Rab7 phosphorylation was blocked by acute TBK1 inhibition with BX-795 (Fig. 2C,D). HeLa cells also showed a drop in Rab7 S72 phosphorylation in response to starvation and restoration of Rab7 phosphorylation in response to amino acid re-feeding (Fig. 2E,F). In support of the functionality of the TBK1-GFP chimeric protein, TBK1 KO HeLa cells that were rescued with TBK1-GFP also showed a similar pattern of reduced Rab7 phosphorylation upon starvation and an increase in response to amino acid re-feeding (Fig. EV1F,G). Comparison of total TBK1 and phospho-TBK1 (S172) in the starved and fed state revealed that amino acid re-feeding increased both total and phospho-TBK1 on purified lysosomes (Fig. 2G–I). However, phospho-TBK1 levels in whole-cell lysates did not increase but instead slightly decreased in response to amino acid re-feeding. This may reflect the presence of only a fraction of the total TBK1 pool at lysosomes and differential regulation of lysosomal versus non-lysosomal pools of TBK1. With respect to Rab7, its total abundance at lysosomes was not regulated by either amino acids or TBK1 KO but phospho-Rab7 (S72) levels in both the total lysate and at lysosomes increased in response to amino acid re-feeding and this effect was abolished in cells lacking TBK1 (Fig. 2G,J,K). Thus, although LRRK1 has also been shown to phosphorylate Rab7 and may explain the Rab7 S72 phosphorylation that persists in TBK1 KO cells, the amino acid-regulated increase in Rab7 phosphorylation is predominantly mediated by TBK1 (Hanafusa et al, 2019).

We next took advantage of the robust TBK1-GFP signal to further characterize regulation of lysosomal TBK1 by changes in amino acid availability. Immunoblot analysis of purified lysosomes revealed that TBK1-GFP lysosome abundance increased when starved cells were re-fed with amino acids and this was paralleled by increases in Rab7-S72 phosphorylation (Fig. 3A–E). The

increase in TBK1-GFP lysosome abundance in response to amino acids was independently observed by confocal immunofluorescence microscopy (Fig. 3F,G).

Our discovery that TBK1 is dynamically recruited to and signals from lysosomes in response to amino acid abundance, led us to perform additional experiments to test whether lysosomes serve as platforms for TBK1 signaling in other contexts. STING is well known to activate TBK1 at the Golgi and to be subsequently degraded within lysosomes (Balka et al, 2023; Gentili et al, 2023; Gonugunta et al, 2017; Kuchitsu et al, 2023). Along with the expected TBK1 and STING phosphorylation, we observed an increase in Rab7 S72 phosphorylation in the cGAMP-treated cells that is consistent with a reported role for TBK1-mediated Rab7 phosphorylation downstream of STING (Fig. EV2A) (Ritter et al, 2020). This STING-dependent Rab7 phosphorylation was abolished in TBK1 KO cells and rescued by the expression of a TBK1-GFP transgene (Fig. EV2A). This may reflect TBK1 activation by STING on late endosomes and lysosomes before its internalization into the lysosome lumen for degradation. These observations pertaining to the relationship between TBK1, STING, and Rab7 show that TBK1 signaling from lysosomes can be regulated by both STING and amino acid availability. However, given that amino acids still elicit Rab7 phosphorylation in STING KO cells (Fig. EV2B), our new observations concerning amino acids as regulators of lysosomal TBK1 are independent from the STING pathway for TBK1 activation.

To determine whether amino acid-regulated changes in TBK1-dependent Rab7 phosphorylation are secondary to mTORC1 activity, we inhibited mTOR with Torin 1 and tested the effect of amino acid starvation-refeeding. We found that even when mTORC1 was strongly inhibited (as revealed by loss of S6K1 phosphorylation), amino acid refeeding still stimulated an increase in Rab7 phosphorylation that was sensitive to TBK1 inhibition (Fig. EV2C). We also observed that depletion of RagC, part of the complex that recruits mTORC1 to lysosomes when amino acids are abundant (Lama-Sherpa et al, 2023), abolished the activation of mTORC1 by amino acids but did not impair TBK1-dependent Rab7 phosphorylation (Fig. EV2D). These results demonstrate that TBK1 and Rab7 phosphorylation changes are not secondary to mTORC1 activation by amino acids. They also argue against a requirement for Rag GTPases and their upstream amino acid

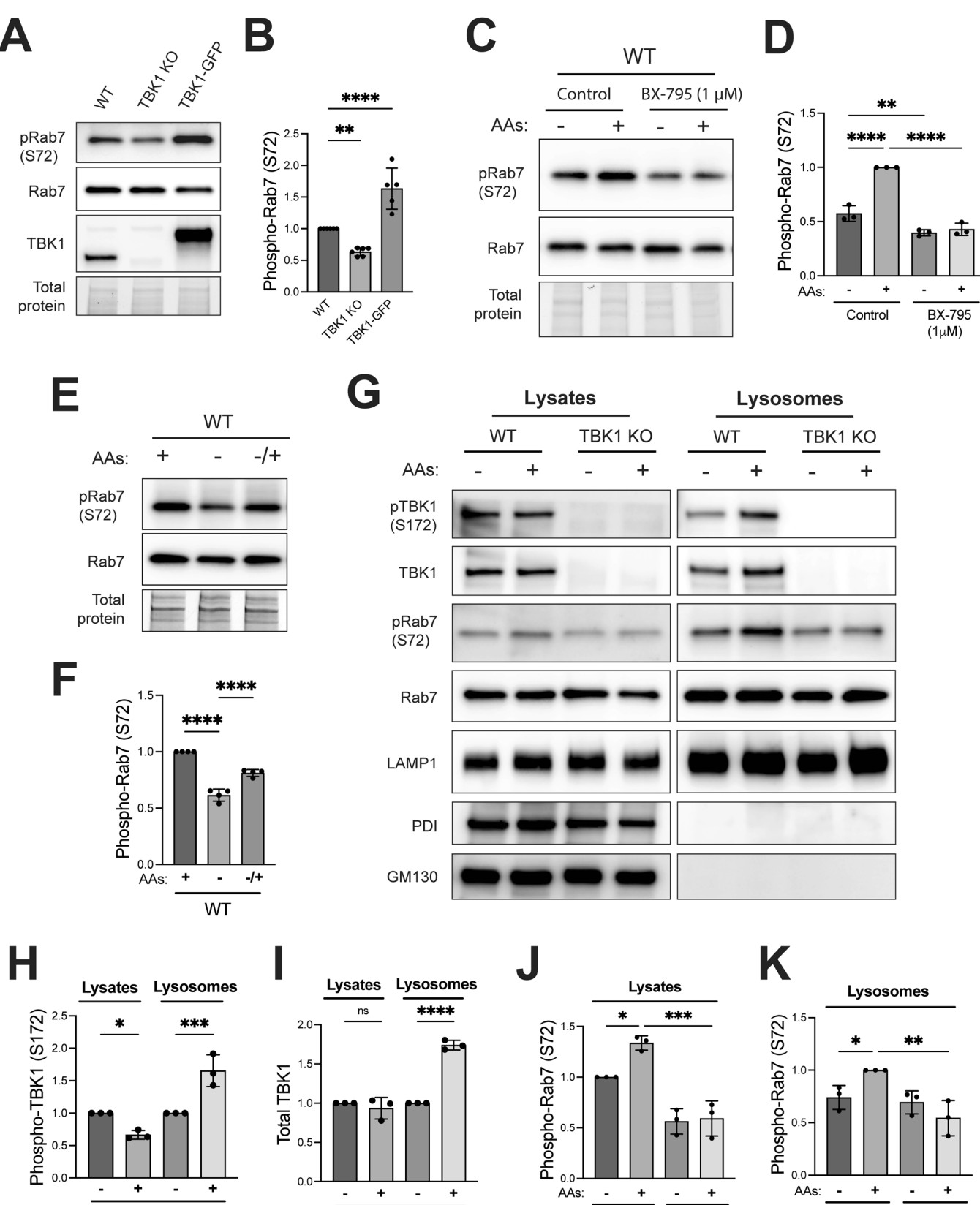

**Figure 2. TBK1-dependent Rab7 phosphorylation is regulated by amino acid availability.**

(A) Immunoblot analysis of WT, TBK1 KO, and TBK1 KO + TBK1-GFP HeLa cells under basal conditions. (B) Phospho-Rab7 (S72) protein levels from panel A were quantified and normalized to total Rab7 (expressed as fold change from WT). Data from 5–6 biological replicates was plotted and statistical significance was determined by ordinary one-way ANOVA with Šidák post hoc (mean ± SD; **$p = 0.0064$; ****$p < 0.0001$). (C) Immunoblot analysis of RAW264.7 cells that underwent amino acid starvation (60 min) and re-feeding (60 min) in the presence of 1 μM BX-795 or 0.01% DMSO (Control). (D) Phospho-Rab7 protein levels were quantified, normalized to total Rab7 and expressed as fold change from re-fed or untreated WT cells. Data reflects three biological replicates and statistical significance was determined by ordinary one-way ANOVA with Šidák post-test ($n = 3$; mean ± SD; ****$p < 0.0001$; **$p = 0.0059$). (E) Immunoblot analysis of pRab7 (S72), total Rab7 in whole-cell lysates of WT HeLa cells under basal conditions (+), starved (−) and amino acid re-fed (−/+). (F) Phospho-Rab7 (S72) protein levels were quantified, normalized to total Rab7 and expressed as fold change from basal conditions. Data reflects four biological replicates and statistical significance was determined by ordinary one-way ANOVA with Šidák post-test ($n = 4$; mean ± SD; ****$p < 0.0001$). Total protein was detected by the TGX stain-free method. (G) Immunoblot analysis of the indicated proteins in cell lysates (Lysates) and magnetically isolated lysosomes (Lysosomes) of WT and TBK1 KO HeLa cells starved for 60 min (−) and re-fed with amino acids for 60 min (+). Blots for Lysate and Lysosome fractions were obtained in parallel from the same membrane. (H, I) Quantification of phosphorylated TBK1 (S172) and total TBK1 normalized to total Rab7 in Lysates and Lysosomes and expressed as fold change compared to starved cells (mean ± SD; *$p = 0.0239$; ***$p = 0.0005$; ****$p < 0.0001$; ns, not significant). (J, K) Phospho-Rab7 (S72) was quantified and normalized to total Rab7 in Lysates and Lysosomes (mean ± SD; *$p = 0.0192$; ***$p = 0.0001$; *$p = 0.0494$; **$p = 0.0026$). Data reflects three biological replicates and statistical significance was determined by ordinary one-way ANOVA with Šidák post-test. Source data are available online for this figure.

sensors in communicating changes in amino acid availability to TBK1 and Rab7. The lack of effect of mTORC1 inhibition on TBK1 activity toward Rab7 also argues against major contributions of autophagy, a process that is activated downstream of mTORC1 inhibition (Goul et al, 2023; Liu and Sabatini, 2020).

## TBK1-dependent Rab7 phosphorylation is important for efficient mTORC1 activation

It was previously reported that Rab7 negatively regulates mTORC1 kinase activity (Kvainickas et al, 2019). However, this study did not investigate a role for Rab7 S72 phosphorylation in mTORC1 regulation. Our results from TBK1 and Rab7 perturbations support of a model wherein TBK1-mediated phosphorylation of Rab7 relieves Rab7-dependent inhibition of mTORC1. Consistent with this model wherein TBK1 acts through Rab7 to control mTORC1, expression of dominant negative Rab7 (T22N) rescued the impaired mTORC1 activation in TBK1 KO cells (Fig. 4A). Likewise, Rab7 KO increased mTORC1 activation and the double KO of Rab7 + TBK1 rescued the mTORC1 activation defect arising from loss of TBK1 (Fig. 4B,C). The observation that mCherry-tagged wild-type Rab7 rescued mTORC1 over-activation in Rab7 KO cells, ensured that the relationship between Rab7 and mTORC1 was not an off-target effect of the genome editing or an effect of the clonal selection that was required to generate the Rab7 KO line (Fig. 4D).

Comparison of Rab7 KO cells that were rescued with equal amounts of wild-type mCherry-Rab7 versus mCherry-Rab7-S72A revealed that this mutant Rab7 that could not be phosphorylated by TBK1 more strongly suppressed mTORC1 activation (Fig. 5A,B). This further supports the model where phosphorylation of Rab7 by TBK1 relieves Rab7-dependent suppression of mTORC1 activity. Interestingly, cells expressing the Rab7-S72A mutant also had significantly less TBK1 at lysosomes (Fig. 5C,D). The concept that phosphorylation on S72 inhibits major functions of Rab7 is also supported by analysis of STING downregulation following its activation by cGAMP. This was impaired in Rab7 KO cells, rescued by re-expression of wild-type Rab7 and the Rab7-S72A mutant caused even stronger agonist-induced STING degradation (Fig. EV3). These effects of Rab7 phosphorylation on STING degradation match expectations raised by a previous study and thus provide validation for the tools that we have applied to amino acid-dependent regulation of TBK1 at lysosomes (Ritter et al, 2020).

## TBK1 also has Rab7-independent effects on lysosomes

Rab7 has multiple roles in the endolysosomal system and changes in lysosome size occur in response to Rab7 perturbations (Bucci et al, 2000; Kummel et al, 2022). We observed that Rab7 KO and TBK1 KO cells both exhibited a similar increase in late endosome/lysosome size (Figs. 1G and EV4A–C). This effect was increased further in Rab7 + TBK1 double KO cells (Fig. EV4A,B). The lysosome size alterations in TBK1 KO cells were on target as they were rescued by re-introducing TBK1-GFP (Fig. EV4C). Meanwhile, both wild type and S72A mutant mCherry-tagged Rab7 proteins localized to LAMP1-positive late endosomes/lysosomes and reversed the increase in lysosome size that occurs in Rab7 KO cells (Fig. EV4D,E). However, this effect was more variable with the S72A mutant. The additive effect of the TBK1 and Rab7 KOs and the partial rescue with the Rab7-S72A mutant indicate that TBK1 likely has additional effects on lysosomes via substrates beyond Rab7.

## TBK1-E696K ALS-FTD mutant exhibits constitutive lysosome localization, elevated Rab7 phosphorylation, and increased mTORC1 activity

TBK1 loss-of-function and missense mutations cause ALS and FTD (Cirulli et al, 2015; Freischmidt et al, 2015; Pottier et al, 2015; Pottier et al, 2019). It was previously shown that the ALS and FTD-associated TBK1-E696K mutant leads to loss-of-function phenotypes related to mitophagy due to reduced interaction with optineurin (Harding et al, 2021; Li et al, 2016; Moore and Holzbaur, 2016; Pottier et al, 2015). However, it is still unknown if this is the only TBK1-dependent process affected by this mutation. To define the impact of the TBK1-E696K mutation on our newly discovered lysosomal functions of TBK1, we expressed GFP-tagged wild type or E696K TBK1 (TBK1-WT and TBK1-E696K) in TBK1 KO HeLa cells. Confocal immunofluorescence experiments revealed that stably expressed TBK1-E696K was much more abundant at lysosomes than wild-type TBK1 (Fig. 6A,B). This was accompanied by more Rab7 S72 phosphorylation (Fig. 6C). This increased activity for the TBK1-E696K mutant was not restricted to just Rab7 as there were also increases in TBK1 autophosphorylation and STING phosphorylation in the TBK1-E696K cells under basal growth conditions

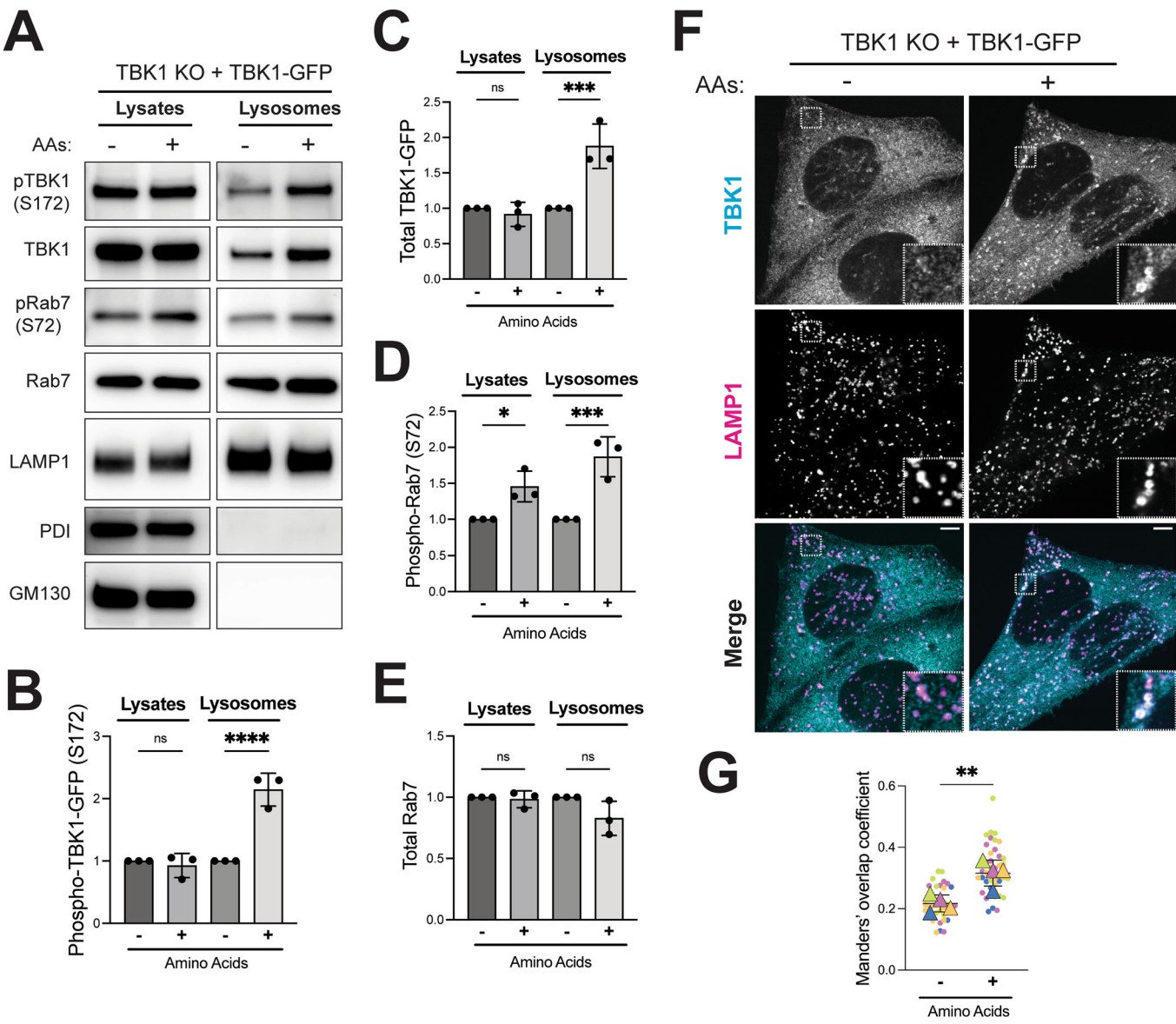

**Figure 3. TBK1 is activated and recruited to lysosomes when amino acids are abundant.**

(A) Immunoblot analysis of the indicated proteins in cell lysates (Lysates) and magnetically isolated lysosomes (Lysosomes) of TBK1 KO HeLa cells that stably express TBK1-GFP that were starved for 60 min (−) and then refed with amino acids (3X MEM) for 30′ (+). (B–E) Quantification of the impact of amino acid re-feeding on abundance of the indicated proteins in whole-cell lysates and isolated lysosomes, normalized to total Rab7 (or in the case of total Rab7 normalized to total LAMP1). Data from three biological replicates were plotted as fold change compared to starved cells. Statistical significance was determined by ordinary one-way ANOVA with Šidák post-test (mean ± SD; ****$p < 0.0001$; ***$p = 0.0006$; *$p = 0.0255$; ***$p = 0.0006$; ns, not significant). (F) Immunofluorescence super-resolution microscopy analysis of TBK1 and LAMP1 in TBK1 KO + TBK1-GFP HeLa cells starved for 60 min (−) and re-fed with amino acids (3X MEM) for 30′ (+). (G) TBK1-LAMP1 colocalization was measured based on Manders' overlap coefficient (M1), data reflects three biological replicates and statistical significance determined by unpaired t test ($n = 3$; mean ± SD; **, two-tailed $p = 0.0078$). Each dot represents a region of interest that contains 1–2 cells as per the representative data in panel (E). Dots are colored to represent data from individual replicates. Triangles summarize the means from each independent experiment. Error bars summarize mean ± standard deviation. Insets are magnified 3X. Scale bar: 5 μm. Source data are available online for this figure.

(Fig. 6C). Increased Rab7 phosphorylation by TBK1-E696K also occurred in the context of amino acid stimulation and was accompanied by more mTORC1 activity (Fig. 6D–G). Taken together these results indicate that TBK1-E696K has gain-of-function properties with respect to lysosome localization, autophosphorylation, Rab7 phosphorylation, mTORC1 activation, and STING phosphorylation.

## Discussion

TBK1 is well characterized for phosphorylating multiple physiologically important substrates related to innate immunity and organelle quality control (Fitzgerald et al, 2003; Harding et al, 2021; Moore and Holzbaur, 2016; Pied et al, 2022; Weidberg and Elazar, 2011). However, two unbiased phospho-proteomics studies

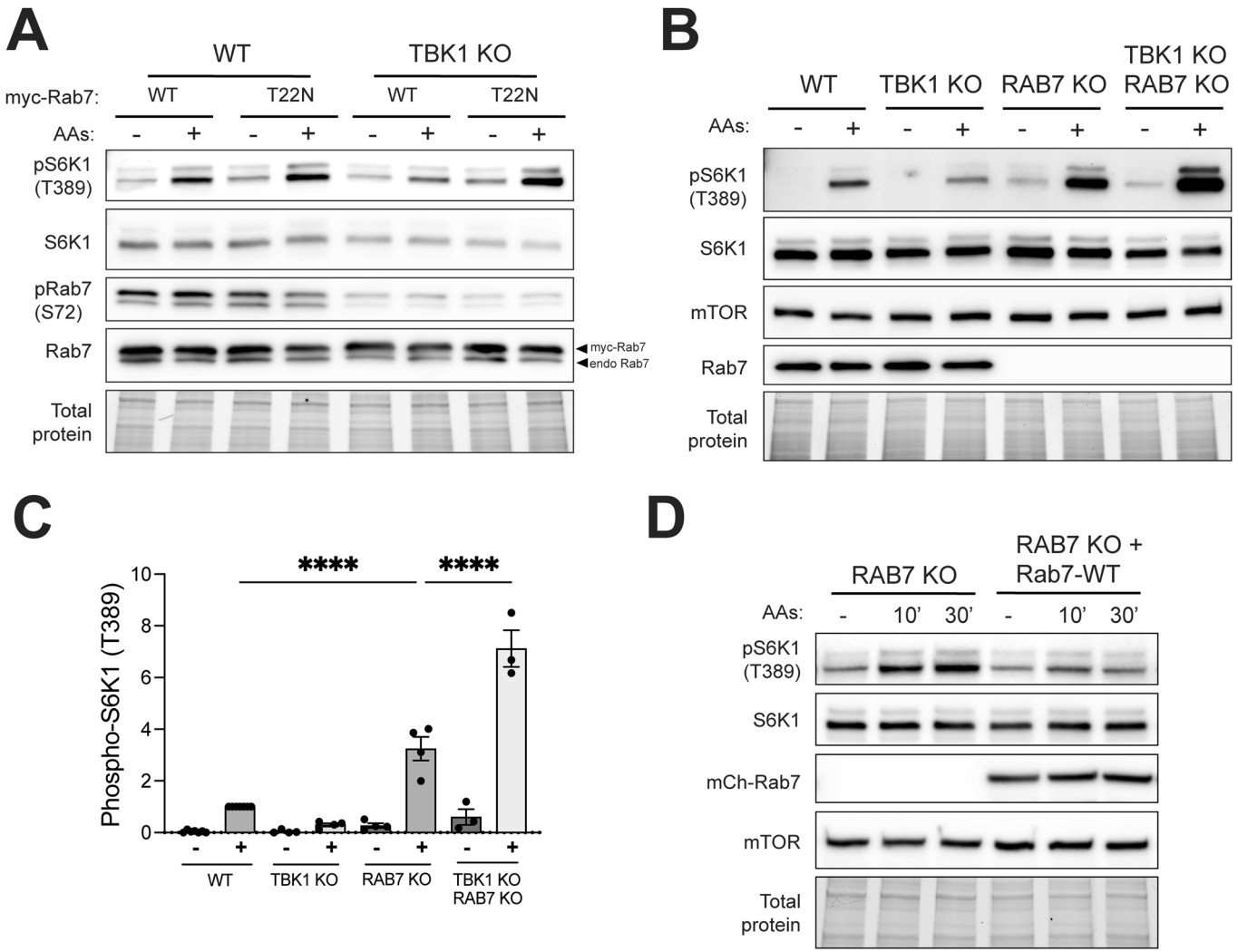

**Figure 4. TBK1-Rab7 axis controls mTORC1 signaling.**

(A) Immunoblotting of the indicated proteins from WT and TBK1 KO HeLa cells transiently expressing myc-tagged wild-type Rab7 or Rab7-T22N starved for 60 min (−) and re-fed with amino acids for 60′ (+). (B) Immunoblotting of the indicated proteins from WT, TBK1 KO, RAB7 KO, and TBK1 KO RAB7 KO HeLa cells starved for 60 min (−) and re-fed with amino acids for 60 min (+). (C) Quantification of pS6K1 (T389) normalized to total S6K1. Plotted data represents fold change compared to WT cells re-fed with amino acids and was derived from at least three biological replicates. Statistical significance was determined by ordinary one-way ANOVA with Šidák post hoc (mean ± SD; ****$p < 0.0001$). (D) Immunoblotting of the indicated proteins of RAB7 KO and RAB7 KO stably expressing mCherry-tagged wild type Rab7 (RAB7-WT) of starved cells for 60 min (−) and re-fed with amino acids for 10 min and 30 min (10′ and 30′). The TGX stain-free method was used to confirm similar total protein levels in each sample. Source data are available online for this figure.

identified Rab7 as a major target for TBK1 that had gone unnoticed in earlier studies of TBK1 signaling (Heo et al, 2018; Ritter et al, 2020). Given that Rab7 localizes to late endosomes and lysosomes this suggested an important but unappreciated endolysosomal role (or roles) for TBK1 and raised questions about both what regulates TBK1 at lysosomes and the physiological consequences of Rab7 phosphorylation. Our results define a nutrient-regulated pathway wherein amino acid availability promotes TBK1 activity at lysosomes. Given that TBK1 was not previously known to signal from intact lysosomes, these results raised questions about what, if anything, it might be doing. Our results show that by phosphorylating Rab7, lysosomal TBK1 relieves inhibition of mTORC1 activation by Rab7. The TBK1-E696K mutant that causes ALS-FTD constitutively localizes to lysosomes, phosphorylates Rab7 and

enhances mTORC1 amino acid responsiveness. Collectively, these results define a sub-population of lysosome-localized TBK1 that is required for efficient activation of mTORC1 when amino acids are abundant. The observation that TBK1-E696K mutant that causes ALS-FTD constitutively localizes to lysosomes, phosphorylates Rab7 and enhances mTORC1 amino acid responsiveness strengthens the link between lysosomal TBK1 and mTORC1 activation and suggests a possible neurodegenerative disease relevance.

Amino acid availability is a major regulator of mTORC1 activation (Lama-Sherpa et al, 2023; Lawrence and Zoncu, 2019; Liu and Sabatini, 2020). This is well appreciated to occur through signals that are sensed and integrated to control the nucleotide loading state of the lysosome-localized Rag GTPase heterodimers that bind mTORC1 and thus control its abundance at lysosomes.

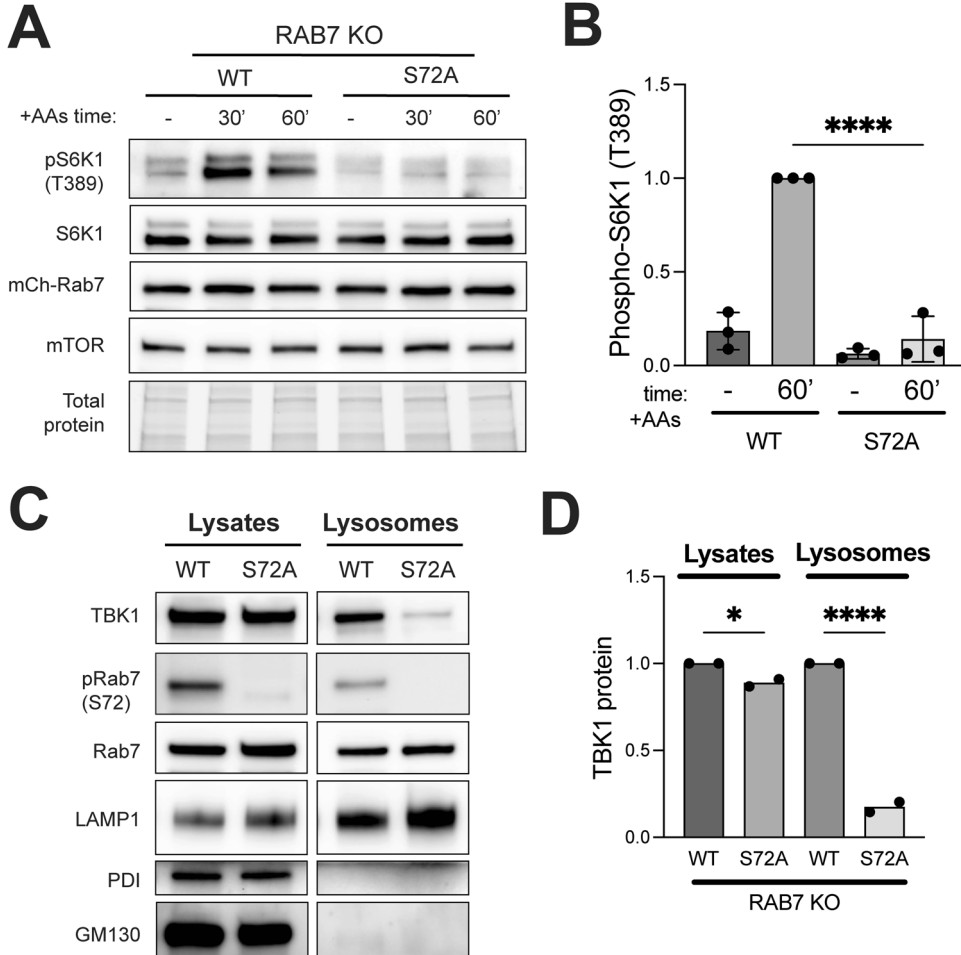

**Figure 5. Rab7-S72 phosphorylation is required for efficient mTORC1 activation and TBK1 lysosome localization.**

(A) Immunoblotting of the indicated proteins from RAB7 KO HeLa cells stably expressing mCherry-tagged wild-type Rab7 (RAB7-WT) or mutant (RAB7-S72A) after cells were starved for 60 min (−) and re-fed with amino acids for 30 min and 60 min (30′ and 60′). (B) Quantification of pS6K1 (T389) normalized to total S6K1 in starved cells (−) versus cells re-fed with amino acids for 60 min (60′). Data from three biological replicates expressed as fold change compared to Rab7-WT re-fed for 60 min and statistical significance determined by ordinary one-way ANOVA with Šidák post hoc test (n = 3; mean ± SD; ****p < 0.0001). TGX stain-free signal was used as a loading control (Total protein). (C) Immunoblot analysis of the indicated proteins in cell lysates (Lysates) and magnetically isolated lysosomes (Lysosomes) of Rab7-WT and Rab7-S72A HeLa cells under basal conditions. (D) TBK1 protein levels were quantified, normalized to Rab7 in the Lysates and Lysosomes and expressed as fold change compared to the re-fed condition (n = 2 biological replicates). Statistical analysis was based on ordinary one-way ANOVA with Šidák post hoc test (mean ± SD; *p = 0.0226; ****p < 0.0001). Source data are available online for this figure.

Our observation that amino acids also act through TBK1 to control mTORC1 signaling (but independent from RagC) thus represents a novel nutrient input into regulation of the mTORC1 signaling pathway. Placing TBK1 in this nutrient-dependent pathway for mTORC1 activation may be of relevance for the anabolic effects of TBK1 in the context of diabetes and obesity (Bodur et al, 2022; Cruz et al, 2018; Oral et al, 2017; Reilly et al, 2013). Identification of specific amino acids, sensors and signal transduction machinery operating upstream of TBK1 is now a priority. Our elucidation of Rab7 as a TBK1 substrate that controls mTORC1 activation joins previous studies that linked TBK1 to mTORC1 activity via phosphorylation of Raptor and mTOR itself (Antonia et al, 2019; Bodur et al, 2018; Ye et al, 2023). The relative contributions of each of these TBK1 substrates to regulation of mTORC1 activity across physiological contexts remains to be determined.

Rab7 is well established as a regulator of multiple aspects of endolysosomal function including membrane traffic, endosome maturation and lysosome subcellular positioning (Kummel et al, 2023). Rab7 was also previously shown to be a negative regulator of mTORC1, an effect that was relieved by TBC1D15, a GTPase activating protein (GAP) for Rab7 (Kvainickas et al, 2019). Our results indicate that Rab7-S72 phosphorylation represents an alternative route to relieving the ability of Rab7 to suppress mTORC1.

Although Rab7-S72 phosphorylation and the functional consequences downstream of it are only partially understood, this post-translational modification was previously linked to regulation of macroautophagy, mitophagy, STING downregulation and EGFR downregulation (Hanafusa et al, 2019; Heo et al, 2018; Ritter et al, 2020; Tudorica et al, 2023). Our results point to a broader role for

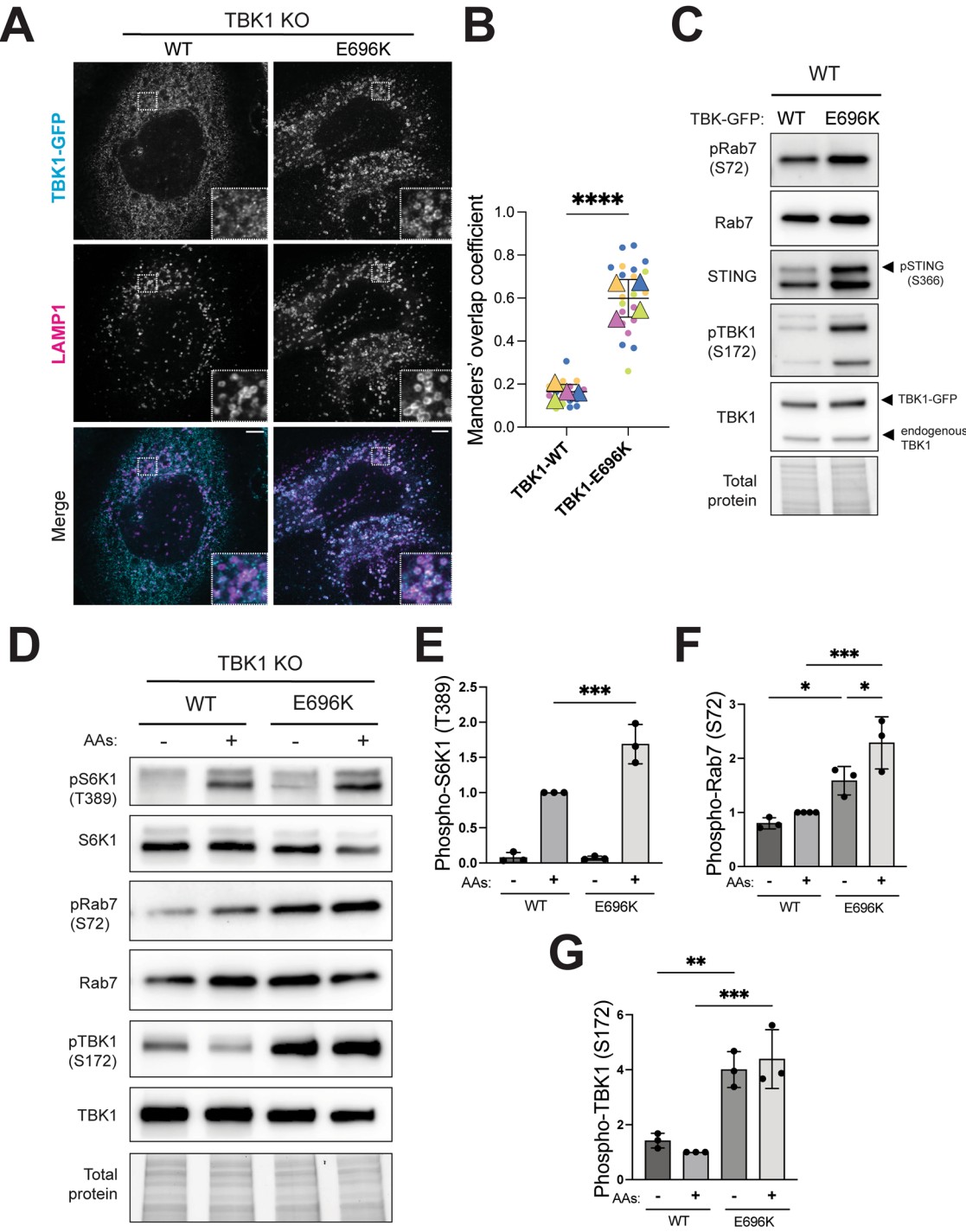

TBK1 as a lysosome regulator. Consistent with the multiple roles for Rab7 in the endolysosomal pathway (Kummel et al, 2023), phosphorylation of S72 in the switch 2 region of Rab7 is likely to affect Rab7-dependent processes by regulating interactions between Rab7 and the effectors that it binds to. Although relatively little is known about Rab7 effectors that differentially respond to phosphorylation on S72 and downstream physiological consequences, some clues have recently emerged. For example, it was reported that Rab7-S72 phosphorylation by LRRK1 increases the

interaction between Rab7 and RILP (Hanafusa et al, 2019). By acting as a scaffold between Rab7 and dynein, RILP promotes the movement of lysosomes toward microtubule minus ends (Jordens et al, 2001). Such movement of lysosomes toward the cell center was previously linked to inhibition of mTORC1 activity (Korolchuk et al, 2011; Pu et al, 2016). However, in our experiments, Rab7 phosphorylation by TBK1 was associated with increased mTORC1 activity and no obvious perinuclear clustering of lysosomes. This suggests the need for additional Rab7 effectors that are sensitive to

**Figure 6. ALS-FTD associated TBK1-E696K mutant accumulates at lysosomes and over-activates mTORC1.**

(A) Immunofluorescence super-resolution microscopy analysis of GFP-tagged wild type (WT) and mutant (E696K) versions of TBK1 stably expressed in TBK1 KO HeLa cells under basal conditions using GFP and LAMP1 antibodies. Scale bar: 5 µm. Insets are magnified 3X. (B) TBK1-LAMP1 colocalization was measured based on Manders' overlap coefficient (M1), data from three biological replicates and statistical significance determined by unpaired t test ($n = 3$; mean ± SD; ****, two-tailed $p < 0.0001$). Each dot represents a region of interest that contains 1–2 cells as per the representative data in panel (A). Dots are colored to represent data from individual replicates. Each dot represents 1–2 cells for a total 30–40 cells over 3 biological replicates. Triangles summarizes the means from each experiment. Error bars show mean ± standard deviation. (C) Immunoblot analysis of the indicated proteins in whole-cell lysates from HeLa cells with transiently expressed TBK1-WT and TBK1-E696K under basal growth conditions. (D) Immunoblotting of the indicated proteins in TBK1 KO HeLa cells that stably express TBK1-WT or TBK1-E696K after starvation for 60 min (−) and amino acid re-feeding for 60 min (+). The TGX stain-free method was used to confirm similar total protein levels in each sample. (E–G) Quantification of phosphorylated proteins normalized to total levels and expressed as a fold change compared to the TBK1-WT cells under re-fed conditions. Plotted data are representative of three biological replicates. Statistical significance based on ordinary one-way ANOVA with Šidák post-test (mean ± SD; ***$p = 0.0004$; *$p = 0.0156$; ***$p = 0.0004$; *$p = 0.0297$; **$p = 0.0023$; ***$p = 0.0004$). Source data are available online for this figure.

S72 phosphorylation. To illustrate the point that phosphorylation-dependent Rab7 effectors are still being discovered, it was very recently shown that phosphorylation of Rab7 differentially controls its interactions with Rubicon and Pacer, two structurally related effectors with distinct roles in autophagy (Tudorica et al, 2023). This strengthens the concept that Rab7 phosphorylation selectively controls effector binding and should motivate the search for additional phospho-selective Rab7 effectors that control mTORC1 and other processes downstream of TBK1 at lysosomes.

The enhanced lysosome localization of the TBK1-E696K mutant provides a valuable tool for investigating the downstream consequences of increasing TBK1 abundance and activity at lysosomes. The constitutive lysosome localization and signaling by the TBK1-E696K mutant also raises the possibility that excessive TBK1 activity at lysosomes contributes to ALS and FTD pathogenesis. To address disease relevance, it would be informative to analyze Rab7 phosphorylation and mTORC1 signaling in samples from patients that carry this mutation. Identifying additional TBK1 mutations that also exhibit increased lysosome localization and Rab7 phosphorylation would also strengthen confidence in the potential disease relevance of the increased TBK1-E696K at lysosomes. However, given that more than 100 TBK1 mutations have so far been discovered in ALS and FTD, it is beyond the scope of this study to systematically test them (Gurfinkel et al, 2022).

Leucine rich repeat kinase 2 (LRRK2) phosphorylates multiple members of the Rab family (but not Rab7) on their switch 2 region at a site that corresponds to S72 in Rab7 (Pfeffer, 2022; Steger et al, 2017; Steger et al, 2016). Phosphorylation on this site modulates the regulatory proteins and effectors that these Rabs bind to (Pfeffer, 2022; Steger et al, 2017). This includes a feed forward relationship between LRRK2 and its ability to both phosphorylate Rab8A and Rab10 and to be activated by recruitment to membranes via interactions with phospho-Rab8A and Rab10 (Vides et al, 2022). Our observations that there was significantly less TBK1 at lysosomes in cells that expressed the Rab7-S72A mutant (Fig. 5C,D) suggests the possibility of a similar feed-forward role for Rab7 phosphorylation in driving further recruitment of TBK1 to lysosomes.

In addition to TBK1, leucine rich repeat kinase 1 (LRRK1) also phosphorylates Rab7 on serine 72 downstream of epidermal growth factor receptor (EGFR) signaling and during mitophagy (Fujita et al, 2022; Hanafusa et al, 2019). This role for LRRK1 likely explains the remaining Rab7 phosphorylation that we observed in TBK1 KO and BX-795 treated cells. It remains to be determined

whether the pools of Rab7 phosphorylated by TBK1 versus LRRK1 are functionally interchangeable. Identification of the relative contributions of each of these mTORC1 inputs in physiological contexts represents an important next step in this field.

In conclusion, we defined an amino acid regulated TBK1 and Rab7-dependent pathway that is required for efficient mTORC1 activation at lysosomes. This establishes TBK1 as a lysosomal protein that is recruited and activated under conditions of amino acid abundance. By identifying a novel subcellular site of action and regulatory input to TBK1, these results open new avenues for investigating the amino acid sensors acting upstream of TBK1. The demonstration that Rab7 phosphorylation relieves inhibitory effects of Rab7 on mTORC1 activation also raises questions about the identity of relevant effectors that are sensitive to Rab7 phosphorylation state. Finally, the potential contributions of lysosomal TBK1 to ALS and/or FTD pathogenesis is another new research direction arising from the foundation laid by this study.

## Methods

### Cell culture, growth conditions, and drug treatments

HeLa (kindly provided by Pietro De Camilli, Yale University) and RAW 264.7 (ATCC) cells were maintained in DMEM + 10% FBS + 1% penicillin/streptomycin (Thermo Fisher Scientific) at 37 °C with 5% $CO_2$. Details of mutant cell lines are summarized in Appendix Table S1. Experiments to test the effect of amino acids on cells were performed as per our previous studies (Amick et al, 2016; Amick et al, 2020; Meng and Ferguson, 2018; Petit et al, 2013). In brief, cells were washed 2× in PBS and incubated in RPMI without amino acids (US Biological) for 1 h and then were shifted to RPMI with MEM Amino Acids (Gibco) for the indicated times and concentrations. BX-795 (Cayman, dissolved in DMSO), cGAMP (Chemietek, 2',3'-cGAMP dissolved in ultrapure water), and Torin-1 (Tocris, dissolved in DMSO) were added to growing cells at the indicated concentrations and compared with cells treated with an equivalent amount of the respective solvents. Details of growth media and drugs are summarized in Appendix Table S2.

For plasmid DNA transfections, $1–3 \times 10^5$ HeLa cells were plated in a 6-well dish per well overnight. The next day, a transfection mix, containing 200 µL OPTiMEM, 6 µL Lipofectamine 2000 (Invitrogen) and 2 µg plasmid, was added to 1.8 mL of fresh new media, per well, and incubated for 24 h.

siRNA transfections were performed with 5 μL of 20 μM RAGC siRNA (On-Target *Plus* Smartpool, Dharmacon) or non-targeting siRNA (NC1 siRNA, IDT), 5 μL RNAiMAX transfection reagent (Invitrogen), and 200 μL OPTiMEM (Invitrogen) that was added to 1.8 mL media containing $1 \times 10^5$ HeLa cells per well in 6-well dish and incubated for 48 h.

Detailed cell culture and transfection protocols can be accessed on protocols.io at: https://doi.org/10.17504/protocols.io.8epv5rrq4g1b/v1 and https://doi.org/10.17504/protocols.io.261ge55byg47/v1, respectively.

## CRISPR/Cas9 genome editing

HeLa TBK1 KO and RAB7A KO cells were created using px459-based Cas9 plasmids. pSpCas9(BB)-2A-Puro (PX459) V2.0 was a gift from Feng Zhang (Addgene plasmid # 62988; http://n2t.net/addgene:62988; RRID:Addgene_62988). Additional plasmid details are presented in Appendix Table S3. The guide RNA sequences to define each target gene are presented in Appendix Table S4. $1 \times 10^5$ cells were plated in a 6-well dish per well and transfection mix, containing OPTiMEM, FuGENE HD, and 1 μg gRNA plasmid, was added to the cells.

## Transposon-mediated genome editing

Transposon-mediated genome editing with piggyBac transposase was used to generate stable HeLa cell lines. $1 \times 10^5$ cells were plated per well in a 6-well dish and transfected with a 3:1 ratio of piggyBac transposon vector with gene-of-interest + piggyBac transposase vector. After 24–48 h, puromycin (Gibco) was added at 2.5 μg/ml. After 24 h of selection, surviving cells were recovered in fresh media without puromycin for 24–48 h and then plated into 96-well dishes at 1 cell per well. Monoclonal cell lines were then confirmed by immunoblotting and fluorescence microscopy. Human TBK1-GFP piggyBac vector was purchased from VectorBuilder. mCh-Rab7A was cloned into a piggyBac plasmid using the NEB HIFI DNA Assembly Kit (NEB) and primers listed in Appendix Table S4. mCherry-Rab7A was a gift from Gia Voeltz (Addgene plasmid #61804; http://n2t.net/addgene:61804; RRID:Addgene_61804). Site-directed mutagenesis of TBK1 and Rab7 genes was performed using Q5 High-Fidelity 2X Master Mix (NEB). Plasmids are listed in Appendix Table S3.

## Immunoblotting

$3 \times 10^5$ HeLa or $1 \times 10^6$ RAW 264.7 cells were seeded per well in a 6-well dish. The following day, cells were washed 2X with cold PBS-1X and scraped in cold lysis buffer [50 mM Tris-HCl pH 7.4, 150 mM NaCl, 1% (v/v) Triton X-100, 1 mM EDTA supplemented with protease and phosphatase inhibitors]. Lysates were centrifuged at $20,817 \times g$ (4 °C) for 8 min to pellet and discard insoluble material. Protein concentrations were measured by the Bradford method (Coomassie Plus Protein Assay Reagent, ThermoFisher Scientific). The lysate supernatants were mixed 1:1 with 2X sample buffer [80 mM Tris-HCl pH 6.8, 3.4 M Glycerol, 3% (w/v) SDS, and 0.02% (w/v) Bromophenol Blue supplemented with fresh 6% (v/v) β-mercaptoethanol] and heated at 95 °C for 3 min.

Protein lysates were electrophoresed in 4–15% miniPROTEAN TGX Stain-Free pre-cast gels (BioRad) using electrophoresis buffer [25 mM Tris Base, 192 mM Glycine, 0.1% (w/v) SDS. Total protein

was visualized through the BioRad TGX stain-free technology with BioRad Chemidoc MP imaging station. After electrophoresis, gels were transferred onto 0.22 μm pore nitrocellulose membrane (Thermo Fisher Scientific) at 100 V for 60 min in transfer buffer (25 mM Tris Base, 192 mM Glycine, 20% (v/v) Methanol). Membranes were blocked in 5% (w/v) non-fat dry milk in TBS-T [10 mM Tris Base, 150 mM NaCl, 0.1% (v/v) Tween-20]. Antibodies were added to the membranes in 5% (w/v) BSA (Sigma-Aldrich) in TBS-T overnight (4 °C) at the indicated concentrations (Appendix Table S5). Membranes were washed 3× for 10 min with TBS-T. Secondary antibodies were added in TBS-T or 5% (w/v) non-fat dry milk TBS-T for 1 h at RT. Membranes were washed 3× for 10 min with TBS-T. Membranes were subjected to chemiluminescence by adding Pico ECL or Femto ECL (Thermo Fisher Scientific) and visualized with a BioRad Chemidoc MP imaging station with care to avoid exposures that resulted in saturated pixels. Quantification of immunoblots was performed with FIJI software version 2.14.0/1.54f (https://fiji.sc; *RRID*:SCR_002285) (Schindelin et al, 2012) by selecting bands with a rectangular selection tool and measuring the intensity for each band. Background subtraction was kept constant within each experiment. Signals of interest were normalized as defined in respective figure legends. A detailed protocol can be accessed on protocols.io at: https://doi.org/10.17504/protocols.io.bp2l6be9zgqe/v1.

## Immunoprecipitation

TBK1-GFP stably expressed in HeLa TBK1 KO cells was immuno-precipitated to analyze TBK1-STING interaction in the presence or absence of STING agonist, cGAMP. Briefly, cells were washed 2X with cold PBS-1X, scraped in ice-cold lysis buffer mentioned above, and centrifuged at $20,817 \times g$ (4 °C) for 8 min. Protein concentrations were measured by the Bradford method and lysates were then incubated with rotation in pre-washed GFP-TRAP beads (Chromotek) for 1 h at 4 °C. Beads were washed 5X with lysis buffer. Proteins were then eluted by 0.5X Laemmli Buffer and boiled at 95 °C for 3 min. A detailed protocol can be accessed on protocols.io at: https://doi.org/10.17504/protocols.io.5jyl822z7l2w/v1.

## Immunofluorescence microscopy

Immunofluorescence microscopy was performed with a Nikon Ti2-E inverted microscope equipped with Spinning Disk Super Resolution by Optical Pixel Reassignment Microscope (Yokogawa CSU-W1 SoRa, Nikon) and Microlens-enhanced Nipkow Disk with pinholes and a 60x SR Plan Apo IR 1.4 NA oil immersion objective. Images were acquired at room temperature. Cells were grown on 12 mm coverslips (GmbH & Co KG), washed in PBS-1X, and fixed by adding 4% PFA (w/v) in 0.1 M sodium phosphate. Cells were fixed at room temperature for 25 min and then washed 3× with PBS-1X. Samples were permeabilized by immersing coverslips in ice-cold methanol for 2–3 s, followed by PBS-1X rinses. Samples were then blocked in PBS-1X with 5% (v/v) normal donkey serum (Jackson) for 1 h at RT. Subsequent antibody incubations were performed in this buffer. Antibodies used in this study are listed in Appendix Table S5.

Images were processed with ImageJ/FIJI version 2.14.0/1.54f (https://fiji.sc; *RRID*:SCR_002285) (Schindelin et al, 2012). For analysis of protein colocalization, Manders' coefficient was

determined with ImageJ/Fiji through the JACoP plugin (https://imagej.net/ij/plugins/track/jacop2.html; RRID:SCR_025164) (Bolte and Cordelieres, 2006). For lysosome size measurements, lysosome perimeter was determined with ImageJ/Fiji with the StarDist plugin (https://imagej.net/plugins/stardist)(Schmidt et al, 2018). Thresholds were equally set for all images analyzed in each experiment. A detailed immunofluorescence protocol can be accessed at: 10.17504/protocols.io.81wgbym7ovpk/v1.

## Lysosome isolation with SPIONs

The magnetic isolation of lysosomes was performed by endocytic loading of cells with superparamagnetic iron oxide nanoparticles (SPIONs) that were prepared as previously described (Amick et al, 2018; Hancock-Cerutti et al, 2022; Rodriguez-Paris et al, 1993). Detailed protocols for the SPION preparation and purification of lysosomes with dextran-conjugated SPIONs can be accessed on protocols.io at: https://doi.org/10.17504/protocols.io.eq2lyn69pvx9/v1 and https://doi.org/10.17504/protocols.io.bp2l61dr1vqe/v1. For cell homogenization, this current study employed a Dounce homogenizer (tight pestle) rather than the Isobiotec homogenizer.

## Statistical analysis

Statistical analysis was carried out with GraphPad Prism version 10.2.2 (www.graphpad.com; RRID:SCR_002798) with specific details about the statistical tests conducted, the number of independent experiments, and $p$ values provided in the corresponding figure legends. GraphPad Prism reports four digits after the decimal point and $P$ values less than 0.0001 are shown as "<0.0001". Graphs (including Superplots for quantification of immunofluorescence experiments) were created by GraphPad Prism version 10.2.2 (Lord et al, 2020). Sample sizes were determined based on recent experience with similar assays. Experiments and their analysis were unblinded.

## Availability of materials

Unique materials described in this study include plasmids and genetically modified cell lines. Plasmids will be made available through Addgene. Cell lines will be available upon request to the corresponding author. Appendix Tables S1–S5 define reagents and their commercial sources (where applicable).

# Data availability

The source data of this paper are collected in the following database record: biostudies:S-SCDT-10_1038-S44318-024-00180-8.

# Peer review information

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

## Acknowledgements

This research was supported by grants from the National Institutes for Health (GM105718) and Aligning Science Across Parkinson's disease (ASAP-000580) through the Michael J. Fox Foundation for Parkinson's Research (MJFF). Agnes Roczniak-Ferguson (Yale) provided valuable feedback and key plasmids. Joe Amick (Yale) generated plasmids for Rab7 knockout. We appreciate constructive feedback from Pietro De Camilli (Yale). We thank Berrak Ugur (Yale) for managing our ASAP project.

## Author contributions

**Gabriel Talaia**: Conceptualization; Resources; Formal analysis; Investigation; Methodology; Writing—original draft; Writing—review and editing. **Amanda Bentley-DeSousa**: Resources; Investigation. **Shawn M Ferguson**: Conceptualization; Data curation; Formal analysis; Supervision; Funding acquisition; Visualization; Writing—original draft; Project administration; Writing—review and editing.

Source data underlying figure panels in this paper may have individual authorship assigned. Where available, figure panel/source data authorship is listed in the following database record: biostudies:S-SCDT-10_1038-S44318-024-00180-8.

## Disclosure and competing interests statement

The authors declare no competing interests.

# Expanded View Figures

**Figure EV1.  TBK1 is required for efficient amino acid-dependent mTORC1 and cGAMP-dependent STING activation.** ▶

(**A**) Immunoblot analysis of phospho-ULK1 at S757, total ULK1, phospho-S6 (S235/S236), and TBK1 of WT and TBK1 KO HeLa cells starved for 60 min (−) and re-fed with amino acids for 60 min (+). (**B**) Quantification of phospho-ULK1 (S757) normalized to total ULK1 and expressed as a fold change compared to the WT cells under re-fed conditions. Data reflets three biological replicates and statistical significance was determined by ordinary one-way analysis of variance (ANOVA) with Šidák *post hoc* test ($n = 3$; mean ± SD; ***$p = 0.0003$). (**C**) Immunoblot analysis of S6K1 phosphorylated at Thr389 [pS6K1 (T389)], total S6K1 and TBK1 in whole-cell lysates from WT versus TBK1 KO RAW 246.7 cells that were starved of amino acids for 60 min (−) and then re-fed with amino acids for 60 min (+). The TGX stain-free method was used to visualize total protein. Asterisk indicates a non-specific band in the phospho-S6K1 blots. (**D**) Quantification of phospho-S6K1 (T389) normalized to total S6K1 and expressed as a fold change compared to the WT cells under re-fed conditions. Data reflets three biological replicates and statistical significance was determined by ordinary one-way ANOVA with Šidák *post hoc* test ($n = 3$; mean ± SD; ***$p = 0.0004$). (**E**) Immunoblot analysis of STING, phospho-TBK1-GFP, and total TBK1-GFP in the lysates (Inputs) and immunoprecipitated TBK1-GFP (IP: GFP) of TBK1 KO HeLa cells stably expressing TBK1-GFP untreated (−) or treated with cGAMP (70 µM) for 120 min (+). (**F**) Immunoblot analysis of pRab7 (S72), total Rab7, and total TBK1-GFP (TBK1) in cell lysates of TBK1 KO + TBK1-GFP HeLa cells under basal fed conditions (+), starved (−) and amino acid re-fed (−/+). (**G**) Phospho-Rab7 levels were quantified and normalized to total Rab7 (basal conditions were considered 1). Statistical significance was determined by ordinary one-way ANOVA with Šidák post-test ($n = 3$; mean ± SD; ****$p < 0.0001$).

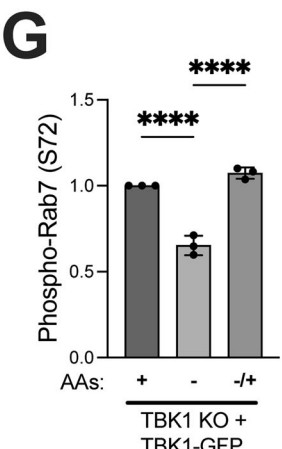

**A**

AAs:   WT    TBK1 KO
       -  +   -  +

pULK1 (S757)

ULK1

pS6 (S235/236)

S6

TBK1

**B**

Phospho-ULK1 (S757)

***

AAs:  -  +   -  +
      WT     TBK1 KO

**C**

              WT      TBK1 KO
                      IKKε KO
AAs:    -   +    -   +

pS6K1 (T389)                    *

S6K1

TBK1

Total protein

**D**

Phospho-S6K1 (T389)

***

AAs:  -  +   -  +
      WT    TBK1 KO
            IKKε KO

**E**

TBK1 KO + TBK1-GFP

          Inputs      IP: GFP
cGAMP:   -    +      -    +

STING

pTBK1-GFP (S172)

TBK1-GFP

SDS-PAGE (TGX-free stain)

**F**

TBK1 KO + TBK1-GFP

AAs:    +    -    -/+

pRab7 (S72)

Rab7

TBK1

Total protein

**G**

Phospho-Rab7 (S72)

****        ****

AAs:   +    -    -/+
      TBK1 KO +
      TBK1-GFP

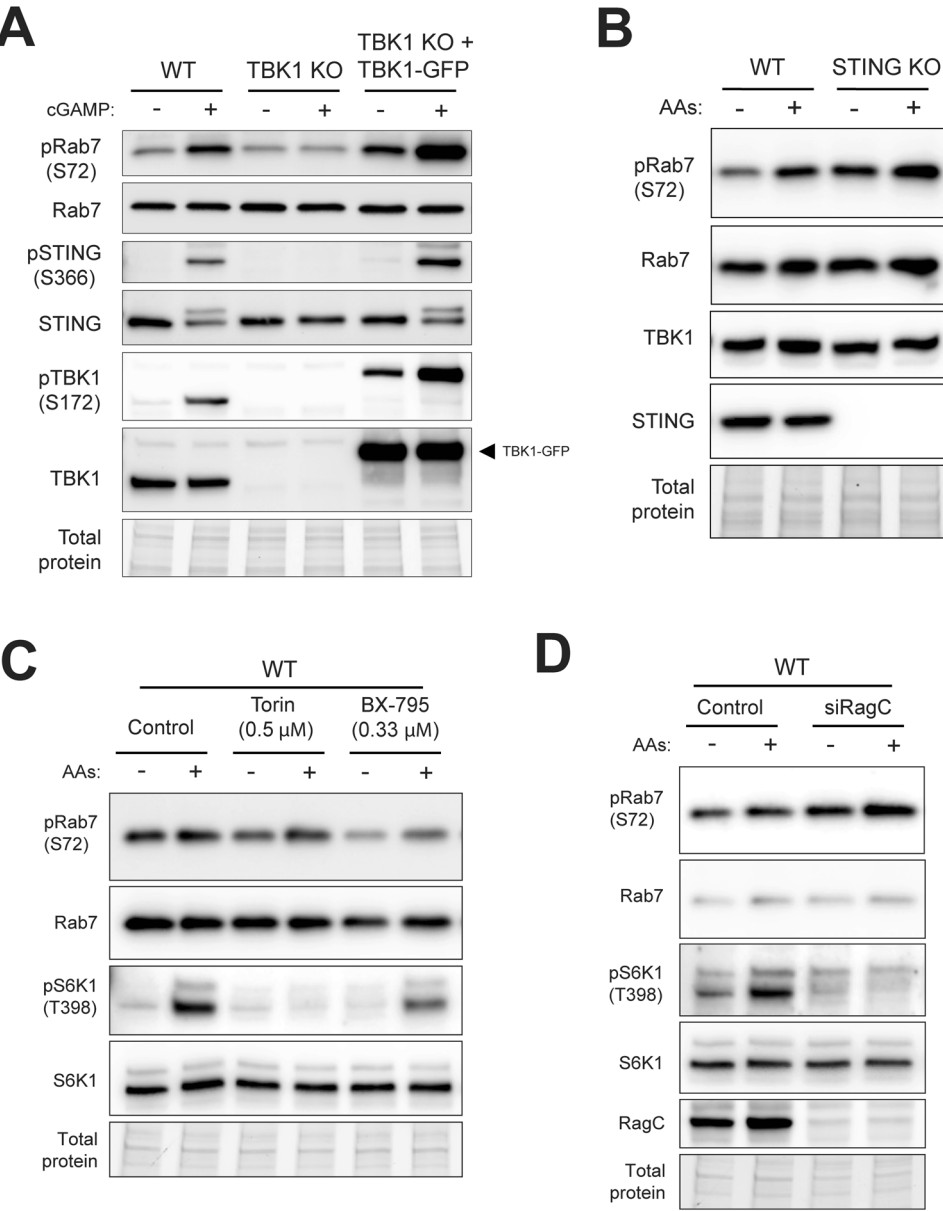

**Figure EV2. STING activation triggers TBK1-mediated Rab7 phosphorylation but inactivation of STING or mTORC1 do not prevent amino acid-dependent TBK1 activity.**

(A) Immunoblot analysis of the indicated proteins in WT, TBK1 KO, and TBK1-GFP HeLa cells untreated (−) or treated with cGAMP (70 µM) for 120 min. (B) Immunoblot analysis of the indicated proteins of WT and STING KO RAW 246.7 cells starved for 60 min (−) and then refed with amino acids for 60 min (+). (C) Immunoblot analysis of the indicated proteins in WT HeLa cells starved for 60 min (−) and re-fed with amino acids for 60 min (+), untreated (Control), DMSO 0.05% (v/v), treated with Torin (0.5 µM), or treated with BX-795 (0.33 µM). (D) Immunoblot analysis of the indicated proteins in WT HeLa cells, transiently transfected with siRNA targeting RAGC (siRagC) or non-targeting/scrambled siRNA (Control), starved for 60 min (−) and re-fed with amino acids for 60 min (+).

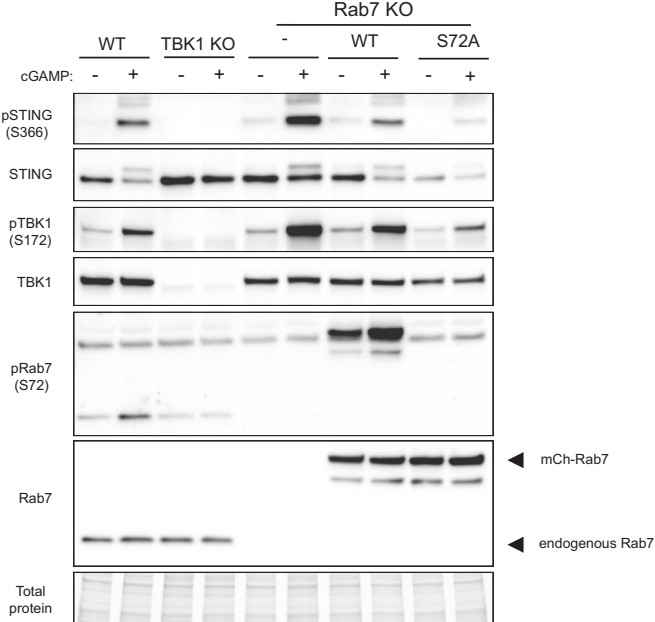

**Figure EV3.  TBK1-mediated Rab7 phosphorylation regulates STING signaling at lysosomes.**

Immunoblot analysis of the indicated proteins of WT, TBK1 KO, Rab7 KO, and Rab7 KO HeLa cells stably expressing mCherry-tagged wild-type or S72A versions of Rab7. These cells were left untreated (−) or were treated with cGAMP (70 µM) for 120 min.

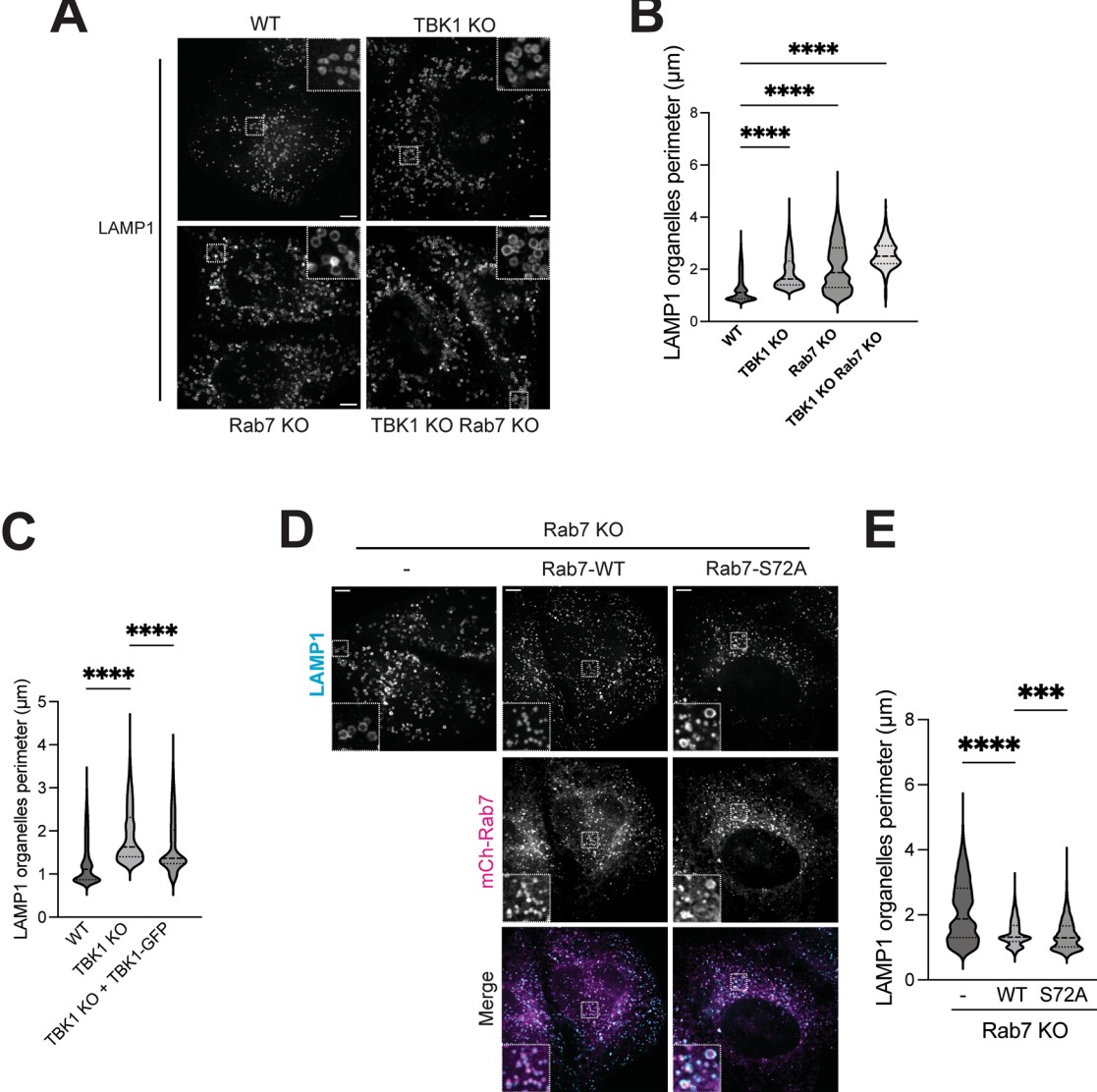

**Figure EV4. TBK1 regulates lysosome size independent of Rab7.**

(A) Immunofluorescence super-resolution spinning disk confocal microscopy analysis of LAMP1 in basal conditions of WT, TBK1 KO, RAB7 KO, TBK1 KO RAB7 KO HeLa cells. (B) Quantification of LAMP1-positive organelle (lysosomes) perimeter (μm) in cells of the indicated genotypes. Scale bar: 5 μm. (C) Quantification of LAMP1-positive organelle (lysosomes) perimeter from WT, TBK1 KO, and TBK1-GFP rescue data presented in Fig. 1G. (D) Immunofluorescence super-resolution microscopy analysis of LAMP1 and mCh-Rab7 in basal conditions for Rab7 KO HeLa cells versus Rab7 KO stably expressing RAB7-WT or RAB7-S72A. (E) Quantification of LAMP1-positive organelle (lysosomes) perimeter (μm) in cells of the indicated genotypes. Data plotted in panels (B), (C) and (E) represents results from 2–4 biological replicates; 11–29 regions of interest; 1–2 cells per region of interest. Statistical significance was determined by the Kruskal–Wallis's test followed by Dunn's post-test (Violin Plot; ****$p < 0.0001$; ***$p = 0.0004$). Scale bar: 5 μm.

