## [Peer Review File · The EMBO Journal]

Lysosomal TBK1 Responds to Amino Acid Availability to Relieve Rab7-Dependent mTORC1 Inhibition

Shawn Michael Ferguson, Gabriel Talaia, and Amanda Bentley-DeSousa

Corresponding author(s): Shawn Michael Ferguson (shawn.ferguson@yale.edu)

Review Timeline:

Submission Date:	29th Feb 24
Editorial Decision:	2nd Apr 24
Revision Received:	7th May 24
Editorial Decision:	16th May 24
Revision Received:	22nd May 24
Accepted:	24th Jun 24

Editor: William Teale

Transaction Report:

This manuscript was transferred to The EMBO JOURNAL following peer review at another journal.

Lysosomal TBK1 Responds to Amino Acid Availability to Relieve Rab7-Dependent mTORC1 Inhibition”

Gabriel Talaia, Amanda Bentley-DeSousa and Shawn M. Ferguson

Response to Reviewers

Reviewer #1 (Remarks to the Author):

This manuscript by Talaia et al described a mechanism where in respond to amino acid, TBK1 inhibits RAB7 which inhibits mTORC1, thereby resulting in activation of mTORC1 at the lysosomes. The authors demonstrated this double negative regulatory mechanism using a variety of experimental designs including KO cells and reconstitutions and confocal microscopy. There data are largely consistent with the authors' conclusions. Since RAB7 is known to inhibit mTORC1 and TBK1 is known to target RAB7, the double negative mechanism described here represent limited advance in my opinion. The double negative mechanism only partially responsible for the mTORC1 response, and a RAB7-independent mechanism remains to be defined. Although it is interesting to see STING activation also promotes TBK1 recruitment to the lysosomes, the physiological relevance to mTORC1 response to amino acid is unclear. How TBK1 naturally travels to the lysosome in respond to amino acid is not defined. The relevance of this mechanism to TBK1-E696K ALS is also interesting, but it is unclear how broad this applies to other ALS TBK1 mutants and how this may fit with other mechanisms of TBK1 already proposed for ALS. Overall, the data is solid but conceptual advance on how this mechanism fit in the broad scope of mTORC1 biology and ALS biology is limited.

We thank the reviewer for recognizing that our data is solid.

As summarized in the Abstract, we have generated multiple new insights that define a role for TBK1 in coordinating lysosomal response to nutrient availability. Key new insights include:

1. The identification of a pool of TBK1 that resides at lysosomes.
2. The abundance and activity of this lysosomal TBK1 is regulated by amino acid availability.
3. Rab7 (serine 72) is a lysosomal target of TBK1.
4. TBK1-dependent phosphorylation of Rab7 relieves the inhibitory effect of Rab7 on mTORC1 signaling from lysosomes.
5. The TBK1-E696K mutant that causes ALS-FTD exhibits constitutive lysosome localization, elevated Rab7 phosphorylation and more mTORC1 activity.

Collectively, this establishes that lysosomes act as a platform for TBK1 signaling. We furthermore show that this lysosomal TBK1 is functionally relevant through analysis of Rab7 phosphorylation and mTORC1 signaling. We agree that additional targets of TBK1 may further contribute to lysosome regulation. We also acknowledge that our new findings raise interesting mechanistic questions about the mechanisms that support TBK1 recruitment to lysosomes and the communication between Rab7 and mTORC1. We believe that our study provides a foundation for future efforts to address such questions.

With respect to the STING-dependent regulation of TBK1 at lysosomes and its relationship to mTORC1, we agree that this may distract from the main message of our paper and have removed it from Figure 3.

Reviewer #2 (Remarks to the Author):

The regulation of mTOR at lysosomes is always of interest. This study provides a hint that the kinase PBK1 by phosphorylating Rab7 controls mTOR activity, as read by phosphorylation of S6K1. This is mainly shown by WBs that are poorly interpretable since the loading control for S6K1 is usually overexposed.

For example Fig 1A lane 2 and 4. In the TBK1 KO +AA case there is indeed less pS6K1 but the S6K1 signal itself is certainly also lower (but as it is over-exposed, this is impossible to judge correctly).

We wish to emphasize that our major contribution is the identification of lysosomes as a platform for TBK1 signaling that is controlled by changes in amino acid availability. Downstream effects on Rab7 and mTORC1 demonstrate the functional relevance of this pool of lysosomal TBK1.

With respect to concerns about the quality of our Western blots, we use the BioRad Chemidoc MP imaging system (<https://www.bio-rad.com/en-us/category/chemidoc-imaging-systems>) for detection of the chemiluminescent signal in our immunoblotting experiments and have been careful not to use images with saturated pixels. However, as the appearance of the current images may reduce confidence in our results for people who use other methods of chemiluminescence detection, we have replaced the following images with shorter exposures:

Fig. 1A (S6K1)
Fig. 1C (Rab7)
Fig. 1E (S6K1)
Fig. 2C (Rab7)
Fig. 2E (Rab7)
Fig. 4A (S6K1)
Fig. 4B (S6K1)
Fig. S1E (Rab7)

The quantification in Fig 1B is then also irrelevant as it should be related to input protein.

The quantification in Fig. 1 B takes into account the abundance of phospho-S6K1 normalized to the total amount of S6K1 in each lane. This strategy is defined in the figure legend. Having addressed the concern about over-exposure (see preceding point), we conclude that we have implemented a reasonable strategy for using S6K1 phosphorylation on T389 as a readout of mTORC1 activity.

These overexposed bands are shown throughout the manuscript for the ref (Rab7 or S6K1) and the quantification may then be incorrectly interpreted and should be related to input S6K1 protein (ie the 'relative amounts of proteins phosphorylated').

We addressed the concern of over-exposure and normalization to total amounts of the protein of interest in our responses to previous points.

The microscopy is used to show that TBK1 can move to membranes labelled for LAMP1 or Rab7. These are of low quality as well and do not show any statistics.

The reason for concern with microscopy quality is puzzling. Immunofluorescence microscopy was performed with a Nikon Ti2-E inverted microscope equipped for Spinning Disk Super Resolution by Optical Pixel Reassignment (Yokogawa CSU-W1 SoRa, Nikon) and a 60x SR Plan Apo IR 1.4 NA oil immersion objective. This has allowed us to generate and present images that detect lysosomes as discrete "donut-like" puncta. This represents the resolution limit of what can be achieved with light microscopy and is more than sufficient to demonstrate changes in TBK1 abundance at lysosomes. We have employed both positive and negative

controls to ensure the specificity of TBK1 detection. To further strengthen confidence in our key discovery of amino acid-regulated recruitment to lysosomes, we have added new quantification of this finding (new Figure 3G). We have corroborated light microscopy observations with quantified immunoblots from purified lysosomes. We have furthermore detected parallel changes in TBK1-dependent phosphorylation of Rab7, a protein of that resides at late endosomes/lysosomes. We have thus provided multiple independent lines of evidence to support our major conclusions about a role for TBK1 at lysosomes.

Fig 1G -for example- is one where TBK1 would colocalize partially with basically any intracellular compartment.

The results in Figure 1G reflect what we observed when we performed immunofluorescence and spinning disk confocal microscopy to detect the subcellular localization of TBK1. This included TBK1 KO cells as a negative control and a rescue line that moderately over-expressed TBK1 as a positive control. These experiments suggested lysosomal localization of TBK1 but were not definitive on their own. We now acknowledge more directly in the text that while these results are consistent with possible TBK1 localization to lysosomes, additional experiments were required to more definitively test this idea. We thus performed additional experiments in subsequent figures to further investigate this topic. This includes immunoblot analysis of purified lysosomes combined with a series of genetic and pharmacological perturbations. Importantly, we demonstrate in Figure 3 that acute re-feeding of starved cells with amino acids results in more robust TBK1 localization to lysosomes and this is apparent by immunofluorescence and immunoblotting approaches (both quantified). Thus, the data in Figure 1 is not presented as the final evidence on this topic, it is just the initial observations that we subsequently developed to a more conclusive outcome. Collectively, our data supports the existence of a pool of TBK1 that undergoes regulated recruitment to lysosomes.

Figure 3E shows a nice and substantial LAMP signal and a pTBK1 signal in the nucleus.

These images were taken at a focal plane near the bottom of the cell that bisects both nucleus and cytoplasm. As a result, lysosomes that reside in the cytoplasm below the nucleus overlay with the nucleus in the images. We do not propose the existence of lysosomes in the nucleus and our result is consistent with both LAMP1 and TBK1 localizing to lysosomes in the cytoplasm.

We acknowledge that the signal-to-noise for the phospho-TBK1 was not as good as for the total TBK1 and LAMP1. We thus removed the phospho-TBK1 images from Figure 3E (now Figure 3F). Fortunately, the immunoblot analysis of purified lysosomes (Figure 2 G and H; Figure 3 A-C) provide a much clearer demonstration of the presence of phospho-TBK1 at lysosomes and its regulation by amino acid availability. The functionality of this lysosomal TBK1 is further supported by Rab7 phosphorylation at lysosomes (Figure 1H, 2G, 2K, 3A, 3D, 5C, 6D and 6F).

How do the authors know that the signal outside the nucleus is specific?

As outlined above, we do not conclude that there is a lysosomal signal in the nucleus. With respect to specificity of the TBK1 immunofluorescence signal, Figure 1G demonstrates that the signal associated with this antibody is absent from TBK1 KO cells. This result is in line with previous independent validation of this antibody (Alshafie et al, F1000Res, 2022). We furthermore independently validated lysosome localization of TBK1 across multiple conditions through immunoblot analysis of isolated lysosomes.

The results are also not quantified unless 2 cells are sufficient nowadays to make an argument...

We did not mean to give the impression that only 2 cells were analyzed. This data was representative of observations from multiple independent experiments. This is now made clear in the legend for Figure 3 and through the new quantification that is presented in the revised panel 3G.

Only the last microscopy image (Figure 6A) is interpretable albeit that it also shows substantial colocalization in the WT case. The observation that the E696 mutant better colocalizes with LAMP1 is likely and(!) is quantified!

We agree that the TBK1-E696K mutant exhibits increased colocalization with LAMP1.

While this is interesting, the quality of the data are relatively poor and not convincing at most places. Microscopy should be performed at much higher resolution and the WBs have to be quantified relative to the input protein levels and should not show overexposed bands.

We have summarized our efforts to address these issues in our previous responses. We are confident that the quality of our immunoblotting and immunofluorescence data in this revised manuscript meets expectations of the cell biology community.

A bar in the microscopy images may help...

We have ensured that all of our microscopy images contain scale bars. The dimensions of the scale bars are defined in the respective Figure Legends.

Reviewer #3 (Remarks to the Author):

In the present study, Talaia et al., presents data on that lysosomal TBK1 regulates mTORC1 through Rab7. The topic is interesting and the manuscript presents high-quality findings, methods, and results in a clear manner. The authors have employed appropriate statistical tests throughout the manuscript to support their conclusions. However, the study leaves major questions unanswered and for some major claims, the direct causal relationships proposed by the authors are not thoroughly substantiated (see below).

We appreciate the recognition of an interesting topic with high quality findings and clear presentation! We recognize challenges in establishing direct causal relationships and have addressed these concerns in a point-by-point manner below.

Major concerns:

1) In their model, the authors propose a sequence of events: amino acids → lysosomal TBK1 activity → Rab7 S72 phosphorylation → lysosomal mTORC1 activity. However, the manuscript lacks mechanistic insights into how Rab7, particularly phosphorylated Rab7, regulates mTORC1 signaling (including mechanism of action, downstream effectors and upstream mediators). In light of this, and considering that both TBK1 and Rab7 can independently impact lysosomal physiology (Fig S4 and PMID: 10679007), it is challenging to determine whether the observed effects on mTORC1 signaling are a direct outcome (i.e., Rab7 directly influencing mTORC1 in some way) or an indirect influence (e.g. Rab7 affecting lysosomal maturation/physiology or other cellular processes that in turn influence mTORC1 signaling). Moreover, the fact that Rab7-

S72A expression regulates TBK1 localization at the lysosomes (Fig 5C) further raises questions about the authors' model.

We recognize the challenges in dissecting the mechanisms whereby Rab7 suppresses mTORC1 signaling. This is an important problem that remains to be conclusively solved. However, we see the main discoveries of our manuscript to be:

- 1) The identification of a pool of TBK1 that undergoes regulated recruitment to lysosomes in response to changes in amino acid availability. This is distinct from previous reports for a role for TBK1 in the lysophagy response to severe lysosome damage.
- 2) Demonstration that Rab7 is a key substrate of this pool of TBK1 and that TBK1-dependent Rab7 phosphorylation is relevant for mTORC1 signaling. The main goal of the mTORC1 activity assays was to establish a functional impact of the lysosomal TBK1 and phospho-Rab7. Our data supports this. We have been careful not to over-reach in making claims about the precise mechanisms that link phospho-Rab7 to mTORC1. The effects of Rab7 phosphorylation on mTORC1 could occur via multiple mechanisms. What's important for now is that the mTORC1 provides a functional readout for the changes to TBK1 and Rab7 at lysosomes.

With respect to the Rab7-S72A mutant, we have shared our observation that it has a major impact on TBK1 localization to lysosomes. In the discussion, we highlight potential parallels to the relationship between LRRK2 and Rab proteins where it has been proposed that Rab10 is both a recruiter of LRRK2 and a LRRK2 substrate.

2) Expanding on the above point, Rab7 is well documented to play a role in both lysosome maturation (PMID: 10679007) and positioning (see references below), both of which can significantly influence mTORC1 signaling. In particular, based on Korolchuk et al., NCB 2011 (PMID: 21394080), changes in Rab7-activity will likely induce changes in lysosome positioning, which in turn affect mTORC1 signaling levels (see e.g PMID 27799357, PMID: 29030394). However, this scenario has not been explored by the authors.

We recognize that there are multiple mechanisms that potentially connect Rab7 to mTORC1. As a direct mechanism, it has been proposed that Rab7 domains compete with Ragulator domains for space on the surface of lysosomes (Kvainickas et al, JCB, 2019, but this study did not investigate Rab7 phosphorylation). However, as the reviewer rightly points out, Rab7 effects on lysosome maturation and positioning also potentially have an impact. We acknowledge this with our results from Fig. S4 which shows changes in lysosome size in TBK1 KO and Rab7 KO cells. We also now mention studies on the relationship between lysosome positioning and mTORC1 signaling in the discussion but note that its conclusions do not completely fit with predictions arising from our results. Although it's good to place our results into this broader context, resolving these complexities is beyond what can be addressed in a single study. As outlined above, our goal was not to define how the balance between Rab7 and phospho-Rab7 regulates mTORC1 but instead to use mTORC1 as a readout to test whether lysosomal TBK1 and Rab7 phosphorylation has measurable downstream consequences at lysosomes. In so doing, we demonstrate a previously unrecognized role for Rab7 phosphorylation upstream of processes that eventually converge on mTORC1. We also acknowledge the likely existence of other targets for TBK1 at lysosomes and show TBK1-dependent changes in lysosome size that are consistent with this idea. We have edited the results and discussion section to make this point clearer.

3) While the authors propose a model involving amino acid signaling, several aspects remain untested. For instance, they have not investigated: i) Whether alterations in TBK1-Rab7

activity/phosphorylation affect basal mTORC1 signaling levels in growing cells, and ii) if starvation of other factors crucial for mTORC1 signaling (e.g., glucose or growth factors) would lead to similar effects on the lysosomal localization of TBK1 and the phosphorylation of Rab7. Furthermore, considering the previously established connections between TBK1 and autophagy (e.g., PMID: 30627666), it remains uncertain whether the proposed pathway is linked to autophagy (e.g. macroautophagy is potently induced by amino acid starvation). Lastly, as noted by the authors, Rab7 S72 is subjected to regulation by various factors, e.g. PTEN (Swapnil Rohidas Shinde et al.). This may indicate that this phosphorylation mark may serve as a broader regulator of Rab7 and conditions that involve alterations in lysosome positioning, such as amino acid starvation (PMID: 21394080).

We have added new data showing that mTORC1 inhibition does not affect the TBK1-dependent phosphorylation of Rab7 that is stimulated by amino acids (Fig. S2 C and D). Given that mTORC1 is the major regulator of autophagy induction downstream of changes in amino acid starvation, the fact that TBK1 and Rab7 are uncoupled from stimuli that induce autophagy argues against a major role for autophagy in regulating TBK1 at lysosomes. Identifying the amino acid sensors and signal transduction upstream of TBK1 is an important new research direction that can build on the foundation that we provide in this study.

We have updated the results section to mention that the lack of effect of mTORC1 inhibition on Rab7 phosphorylation argues against a major contribution from autophagy.

Our response to the previous point covers the topic of amino acid sensing and lysosome sub-cellular positioning.

4) Is the proposed amino acid signaling pathway involving known amino acid-signaling components, including Rag-GTPases and GCN2?

We have observed that RagC is not required for TBK1-mediated Rab7 phosphorylation by amino acids. This new data is included in Figure S2D. We furthermore discuss the possibility of novel amino acid sensing mechanisms. With respect to GCN2, we are not aware of robust evidence linking it to amino acid sensing at lysosomes. While this does not exclude it as a candidate, further investigations in this direction would require significant new efforts with uncertain outcomes.

Other concerns:

1) In some blots, it appears like only the p70 isoform of S6K is phosphorylated/dephosphorylated (e.g. Fig 1C). In the light of this, it would be good to test additional substrates of mTORC1, like p-4EBP.

In Figure 1C, there is a non-specific band for S6K1 in mouse cells that may be causing confusion. We have added an asterisk to the figure and made note of this in the figure legend.

Due to greater robustness of the p70 signal compared to the p85 signal in the cells that we study, we have focused on it.

With respect to other substrates, in Figure S1A, we show that ULK1 phosphorylation is also sensitive to TBK1 KO. We have added new quantification of this result as panel S1B. This supports the conclusion that the effects of TBK1 inhibition can be generalized to multiple mTORC1 substrates.

2) In Figure 1C and 1E, the Rab7 blots appear overexposed. Consequently, it becomes hard to discern whether the observed decrease in p-Rab7 S72 in Figure 1E is due to a genuine reduction in p-Rab7 S72 or a result of decreased total Rab7 levels (Figure 1E).

We were careful to avoid exposures with saturated pixels. This is readily detectable with the Chemidoc MP imaging platform that we use for our immunoblots.

Nonetheless, to increase confidence in our results, we have updated the following blots to address this concern:

Fig. 1A (S6K1)
Fig. 1C (Rab7)
Fig. 1E (S6K1)
Fig. 2C (Rab7)
Fig. 2E (Rab7)
Fig. 4A (S6K1)
Fig. 4B (S6K1)
Fig. S1E (Rab7)

3) The manuscript lacks details on the methodology used for immunoblotting quantifications. While most of the quantifications appear to align well with the corresponding blots, in Figure 2D, the reported two-fold increase in p-Rab7 S72 does not seem consistent, although this discrepancy may be attributed to the previously mentioned issue of overexposed Rab7 blots.

This result reflects outcome of 3 independent experiments. We have addressed the issue of over-exposed blots (see above). This includes adding an image from a lower exposure for Rab7 in Figure 2C. We have also added extra information to our methods section to help clarify our strategy for immunoblot quantification.

4) In the result section, the authors state regarding ULK and S6K “These two proteins are well-established mTORC1 substrates whose phosphorylation is regulated by amino acid availability (Liu and Sabatini, 2020)”. However, this may come across to a reader that these mTORC1 substrates are exclusively influenced by amino acid availability, which is not accurate. This could be clarified.

In order to address this concern, we now simply refer to ULK1 and S6K1 as direct substrates of mTORC1.

Dear Shawn,

Thank you again for submitting your manuscript to The EMBO Journal. We sent this revised version to two of the three original referees, and have now received reports from both of them, which I have included below. As you will see, referee 2 is satisfied, but referee 1 maintains that some of the data presentation and analysis needs attention. I suggest we meet online to go through the figures together and work out what can be done. Please let me know whether you would be available this week.

Best wishes,

William

William Teale, PhD
Editor
The EMBO Journal
w.teale@embojournal.org

We realize that it is difficult to revise to a specific deadline. In the interest of protecting the conceptual advance provided by the work, we recommend a revision within 3 months (1st Jul 2024). Please discuss the revision progress ahead of this time with the editor if you require more time to complete the revisions. Use the link below to submit your revision:

Referee #1:

The authors promised in the rebuttal that the Figures are improved in a way that the WBs are interpretable. I sadly disagree. It should not be too difficult to provide WBs such that the bands are shown at a grey rather than superblack scale. Now it is impossible to judge the results from the quantifications on the basis of the WBs provided. For example Fig 1E shows that the second lane Rab7 is less than in Lane 1 and 3, but how much is impossible to judge. The quantification in Figure 1F is only possible if the differences between the 4 experiments are super small to make this significant. I have doubts about this. Anyway, the authors simply do not provide compelling data as long as the WBs are shown with such overexposed bands. The microscopy images are still of relative poor quality. The number of LAMP1 lysosomes are excessive and (Figure 6) in the nucleus. It should not be too difficult to make better images that convince this reviewer (rather than arguments in the rebuttal). It would also be good to repeat the experiments in another cell line to show that the observations are more generally correct. Finally, the experiments suggest that first mTOR has to be activated before the TBK1 comes in to phosphorylate Rab7 and not the way suggested by the authors. Small point: molecular weight markers in WBs are also standard.

Referee #2:

The authors have adequately addressed the critiques I previously raised. While the manuscript's inherent weaknesses persist in establishing causal links, these issues are clearly acknowledged and discussed throughout the text

The authors addressed all the issues raised.

Dear Shawn,

Thank you for submitting your revised manuscript to The EMBO Journal. Reviewer #2 raised technical concerns over the quantification of your western blot bands and microscope images. I have therefore asked our data integrity analyst to assess these issues; after this process we have found no points of concern. Before I can finally accept the manuscript though, there are some remaining editorial points which need to be addressed. In this regard would you please:

- complete the author checklists and source data checklists that are associated with this manuscript,
- upload main figures as individual high resolution figure files, removing the black frame
- add up to five keywords,
- provide a valid email address for co-author Amanda Bentley de Sousa
- use the following text in the Data Availability section 'The source data of this paper are collected in the following database record: (add link)',
- include acknowledgement of funding from Michael J. Fox Foundation for Parkinson's Research (MJFF) in our online submission system,
- remove the AC/CrediT section from the text,
- rename the conflict of interest statement as the "Disclosure and competing interests statement",
- in the reference list, limit the number of authors to 10 author names using 'et al.' if this number is exceeded,
- add a table of contents with page numbers to appendix 1,
- correct nomenclature to "Appendix Figure S1" etc. and "Appendix Table S1" etc.,
- remove headings "Supplemental figures" and "Supplemental Methods",
- correct callouts for Table 3 and Table 5,
- reorder manuscript sections and correct headings as follows: Abstract, Introduction, Results, Discussion, Methods, Acknowledgements, Disclosure and competing interests statement, References, Figure legends, Tables and their legends, Expanded View Figure legends,
- provide exact p values in the legends of figures 1b, d, f; 2b, d, f, h-k; 3b-e, g; 4c; 5b, d; 6b, e-g, supplementary figures 1b, d, g; 4bc, e,
- define n in the legend of figure 5b,
- remove standard deviation calculation from figure 5d as n=2. Instead, if you wish, you may state that the bar shown indicates the range of measurement, and
- describe the nature of entity for 'n' in the legends of figures 1b, d; 2d, f; 5d, and supplementary figures 1b, d, g.

We include a synopsis of the paper (see <http://emboj.embopress.org/>). Please provide me with a general summary image, two-sentence summary statement and 3-5 bullet points that capture the key findings of the paper.

I am looking forward to receiving your revised manuscript.

EMBO Press is an editorially independent publishing platform for the development of EMBO scientific publications.

Best wishes,

William

William Teale, PhD
Editor
The EMBO Journal
w.teale@embojournal.org

- a point-by-point response to the referees' comments, with a detailed description of the changes made (as a word file).
- a word file of the manuscript text.
- individual production quality figure files (one file per figure)

- a complete author checklist, which you can download from our author guidelines (<https://www.embopress.org/page/journal/14602075/authorguide>).

- Expanded View files (replacing Supplementary Information)

We realize that it is difficult to revise to a specific deadline. In the interest of protecting the conceptual advance provided by the work, we recommend a revision within 3 months (14th Aug 2024). Please discuss the revision progress ahead of this time with the editor if you require more time to complete the revisions. Use the link below to submit your revision:

All editorial and formatting issues were resolved by the authors.

Dear Shawn,

I am pleased to inform you that your manuscript has been accepted for publication in the EMBO Journal.

Congratulations! I'm really please to publish this study in The EMBO Journal.

Best wishes,

William

William Teale, PhD
Editor
The EMBO Journal
w.teale@embojournal.org
